# Greedy Actor-Critic: A New Conditional Cross-Entropy Method for Policy Improvement

**Samuel Neumann, Sungsu Lim, Ajin Joseph, Yangchen Pan, Adam White, Martha White**
Department of Computing Science
University of Alberta
Edmonton, Alberta, Canada
`{sfneuman,amw8,whitem}@ualberta.ca`

## Abstract

Many policy gradient methods are variants of Actor-Critic (AC), where a value function (critic) is learned to facilitate updating the parameterized policy (actor). The update to the actor involves a log-likelihood update weighted by the action-values, with the addition of entropy regularization for soft variants. In this work, we explore an alternative update for the actor, based on an extension of the cross entropy method (CEM) to condition on inputs (states). The idea is to start with a broader policy and slowly concentrate around maximally valued actions, using a maximum likelihood update towards actions in the top percentile per state. The speed of this concentration is controlled by a proposal policy, that concentrates at a slower rate than the actor. We first provide a policy improvement result in an idealized setting, and then prove that our conditional CEM (CCEM) strategy tracks a CEM update per state, even with changing action-values. We empirically show that our GreedyAC algorithm, that uses CCEM for the actor update, performs better than Soft Actor-Critic and is much less sensitive to entropy-regularization.

## 1 Introduction

Many policy optimization strategies update the policy towards the Boltzmann policy. This strategy became popularized by Soft Q-Learning (Haarnoja et al., 2017) and Soft Actor-Critic (SAC) (Haarnoja et al., 2018a), but has a long history in reinforcement learning (Kober & Peters, 2008; Neumann, 2011). In fact, recent work (Vieillard et al., 2020a; Chan et al., 2021) has highlighted that an even broader variety of policy optimization methods can be seen as optimizing either a forward or reverse KL divergence to the Boltzmann policy, as in SAC. In fact, even the original Actor-Critic (AC) update (Sutton, 1984) can be seen as optimizing a reverse KL divergence, with zero-entropy.

The use of the Boltzmann policy underlies many methods for good reason: it guarantees policy improvement (Haarnoja et al., 2018a). More specifically, this is the case when learning entropy-regularized action-values $Q_\tau^\pi$ for a policy $\pi$ with regularization parameter $\tau > 0$. The Boltzmann policy for a state is proportional to $\exp(Q_\tau^\pi(s, a)\tau^{-1})$. The level of emphasis on high-valued actions is controlled by $\tau$: the higher the magnitude of the entropy level (larger $\tau$), the less the probabilities in the Boltzmann policy are peaked around maximally valued actions.

This choice, however, has several limitations. The policy improvement guarantee is for the entropy-regularized MDP, rather than the original MDP. Entropy regularization is used to encourage exploration (Ziebart et al., 2008; Mei et al., 2019) and improve the optimization surface (Ahmed et al., 2019; Shani et al., 2020), resulting in a trade-off between improving the learning process and converging to the optimal policy. Additionally, SAC and other methods are well-known to be sensitive to the entropy regularization parameter (Pourchot & Sigaud, 2019). Prior work has explored optimizing entropy during learning (Haarnoja et al., 2018b), however, this optimization introduces yet another hyperparameter to tune, and this approach may be less performant than a simple grid search (see Appendix D). It is reasonable to investigate alternative policy improvement approaches that could potentially improve our actor-critic algorithms.

---

Code available at `https://github.com/samuelfneumann/GreedyAC`.

In this work we propose a new greedification strategy towards this goal. The basic idea is to iteratively take the top percentile of actions, ranked according to the learned action-values. The procedure slowly concentrates on the maximal action(s), across states, for the given action-values. The update itself is simple: $N \in \mathbb{N}$ actions are sampled according to a *proposal policy*, the actions are sorted based on the magnitude of the action-values, and the policy is updated to increase the probability of the $\lceil \rho N \rceil$ maximally valued actions for $\rho \in (0, 1)$. We call this algorithm for the actor Conditional CEM (CCEM), because it is an extension of the well-known Cross-Entropy Method (CEM) (Rubinstein, 1999) to condition on inputs[1]. We leverage theory for CEM to validate that our algorithm concentrates on maximally valued actions across states over time. We introduce GreedyAC, a new AC algorithm that uses CCEM for the actor.

GreedyAC has several advantages over using Boltzmann greedification. First, we show that our new greedification operator ensures a policy improvement for the original MDP, rather than a different entropy-regularized MDP. Second, we can still leverage entropy to prevent policy collapse, but only incorporate it into the proposal policy. This ensures the agent considers potentially optimal actions for longer, but does not skew the actor. In fact, it is possible to decouple the role of entropy for exploration and policy collapse within GreedyAC: the actor could have a small amount of entropy to encourage exploration, and the proposal policy a higher level of entropy to avoid policy collapse. Potentially because of this decoupling, we find that GreedyAC is much less sensitive to the choice of entropy regularizer, as compared to SAC. This design of the algorithm may help it avoid getting stuck in a locally optimal action, and empirical evidence for CEM suggests it can be quite effective for this purpose (Rubinstein & Kroese, 2004). In addition to our theoretical support for CCEM, we provide an empirical investigation comparing GreedyAC, SAC, and a vanilla AC, highlighting that GreedyAC performs consistently well, even in problems like the Mujoco environment Swimmer and pixel-based control where SAC performs poorly.

## 2 BACKGROUND AND PROBLEM FORMULATION

The interaction between the agent and environment is formalized by a Markov decision process $(\mathcal{S}, \mathcal{A}, \mathcal{P}, \mathcal{R}, \gamma)$, where $\mathcal{S}$ is the state space, $\mathcal{A}$ is the action space, $\mathcal{P} : \mathcal{S} \times \mathcal{A} \times \mathcal{S} \to [0, \infty)$ is the one-step state transition dynamics, $\mathcal{R} : \mathcal{S} \times \mathcal{A} \times \mathcal{S} \to \mathbb{R}$ is the reward function, and $\gamma \in [0, 1]$ is the discount rate. We assume an episodic problem setting, where the start state $S_0 \sim d_0$ for start state distribution $d_0 : \mathcal{S} \to [0, \infty)$ and the length of the episode $T$ is random, depending on when the agent reaches termination. At each discrete timestep $t = 1, 2, \ldots, T$, the agent finds itself in some state $S_t$ and selects an action $A_t$ drawn from its stochastic policy $\pi : \mathcal{S} \times \mathcal{A} \to [0, \infty)$. The agent then transitions to state $S_{t+1}$ according to $\mathcal{P}$ and observes a scalar reward $R_{t+1} \doteq \mathcal{R}(S_t, A_t, S_{t+1})$.

For a parameterized policy $\pi_{\mathbf{w}}$ with parameters $\mathbf{w}$, the agent attempts to maximize the objective $J(\mathbf{w}) = \mathbb{E}_{\pi_{\mathbf{w}}}[\sum_{t=0}^{T} \gamma^t R_{t+1}]$, where the expectation is according to start state distribution $d_0$, transition dynamics $\mathcal{P}$, and policy $\pi_{\mathbf{w}}$. Policy gradient methods, like REINFORCE (Williams, 1992), attempt to obtain (unbiased) estimates of the gradient of this objective to directly update the policy.

The difficulty is that the policy gradient is expensive to sample, because it requires sampling return trajectories from states sampled from the visitation distribution under $\pi_{\mathbf{w}}$, as per the policy gradient theorem (Sutton et al., 1999). Theory papers analyze such an idealized algorithm (Kakade & Langford, 2002; Agarwal et al., 2021), but in practice this strategy is rarely used. Instead, it is much more common to (a) ignore bias in the state distribution (Thomas, 2014; Imani et al., 2018; Nota & Thomas, 2020) and (b) use biased estimates of the return, in the form of a value function critic. The action-value function $Q^{\pi}(s, a) \doteq \mathbb{E}_{\pi}[\sum_{k=1}^{T-t} \gamma^t R_{t+k} | S_t = s, A_t = a]$ is the expected return from a given state and action, when following policy $\pi$. Many PG methods—specifically variants of Actor-Critic—estimate these action-values with parameterized $Q_{\theta}(s, a)$, to use the update $Q_{\theta}(s, a) \nabla \ln \pi_{\mathbf{w}}(a|s)$ or one with a baseline $[Q_{\theta}(s, a) - V(s)] \nabla \ln \pi_{\mathbf{w}}(a|s)$ where the value function $V(s)$ is also typically learned. The state $s$ is sampled from a replay buffer, and $a \sim \pi_{\mathbf{w}}(\cdot|s)$, for the update.

---

[1] CEM has been used for policy optimization, but for two very different purposes. It has been used to directly optimize the policy gradient objective (Mannor et al., 2003; Szita & Lörincz, 2006). CEM has also been used to solve for the maximal action—running CEM each time we want to find $max'_a Q(S', a')$—for an algorithm called QT-Opt (Kalashnikov et al., 2018). A follow-up algorithm adds an explicit deterministic policy to minimize a squared error to this maximal action (Simmons-Edler et al., 2019) and another updates the actor with this action rather than the on-policy action (Shao et al., 2022). We do not directly use CEM, but rather extend the idea underlying CEM to provide a new policy update.

There has been a flurry of work, and success, pursuing this path, including methods such as OffPAC (Degris et al., 2012b), SAC (Haarnoja et al., 2018a), SQL (Haarnoja et al., 2017), TRPO (Schulman et al., 2015) and many other variants of related ideas (Peters et al., 2010; Silver et al., 2014; Schulman et al., 2016; Lillicrap et al., 2016; Wang et al., 2017; Gu et al., 2017; Schulman et al., 2017; Abdolmaleki et al., 2018; Mei et al., 2019; Vieillard et al., 2020b). Following close behind are unification results that make sense of this flurry of work (Tomar et al., 2020; Vieillard et al., 2020a; Chan et al., 2021; Lazić et al., 2021). They highlight that many methods include a mirror descent component—to minimize KL to the most recent policy—and an entropy-regularization component (Vieillard et al., 2020a). In particular, these methods are better thought of as (approximate) policy iteration approaches that update towards the Boltzmann policy, in some cases using a mirror descent update. The Boltzmann policy $\mathcal{B}_\tau Q(s, a)$ for a given $Q$ is

$$\mathcal{B}_\tau Q(s, a) = \frac{\exp(Q(s, a)\tau^{-1})}{\int_{\mathcal{A}} \exp(Q(s, b)\tau^{-1})db} \tag{1}$$

for entropy parameter $\tau$. As $\tau \to 0$, this policy puts all weight on greedy actions. As $\tau \to \infty$, all actions are weighted uniformly. This policy could be directly used as the new greedy policy. However, because it is expensive to sample from $\mathcal{B}_\tau Q(s, a)$, typically a parameterized policy $\pi_{\mathbf{w}}$ is learned to approximate $\mathcal{B}_\tau Q(s, a)$, by minimizing a KL divergence. As the entropy goes to zero, we get an unregularized update that corresponds to the vanilla AC update (Chan et al., 2021).

## 3 CONDITIONAL CEM

Though using the Boltzmann policy has been successful, it does have some limitations. The primary limitation is that it is sensitive to the choice of entropy (Pourchot & Sigaud, 2019; Chan et al., 2021). A natural question is what other strategies we can use for this greedification step in these approximate policy iteration algorithms, and how they compare to this common approach. We propose and motivate a new approach in this section, and then focus the paper on providing insight into its benefits and limitations, in contrast to using the Boltzmann policy.

Let us motivate our approach, by describing the well-known global optimization algorithm called the Cross Entropy Method (CEM) (Rubinstein, 1999). Global optimization strategies are designed to find the global optimum of a general function $f(\beta)$ for some parameters $\beta$. For example, for parameters $\beta$ of a neural network, $f$ may be the loss function on a sample of data. An advantage of these methods is that they do not rely on gradient-based strategies, which are prone to getting stuck in local optima. Instead, they use randomized search strategies, that have optimality guarantees in some settings (Hu et al., 2012) and have been shown to be effective in practice (Peters & Schaal, 2007; Hansen et al., 2003; Szita & Lörincz, 2006; Salimans et al., 2017).

CEM maintains a distribution $p(\beta)$ over parameters $\beta$, iteratively narrowing the range of plausible solutions. The algorithm maintain a current threshold $f_t$, that slowly increases over time as it narrows on the maximal $\beta$. On iteration $t$, $N$ parameter vectors $\beta_1, \ldots, \beta_N$ are sample from $p_t$; the algorithm only keeps $\beta_1^*, \ldots, \beta_h^*$ where $f(\beta_i^*) \geq f_t$ and discards the rest. The KL divergence is reduced between $p_t$ and this empirical distribution $\hat{I} = \{\beta_1^*, \ldots, \beta_h^*\}$, for $h \leq N$. This step corresponds to increasing the likelihood of the $\beta$ in the set $\hat{I}$. Iteratively, the distribution over parameters $p_t$ narrows around $\beta$ with higher values under $f$. To make it more likely to find the global optimum, the initial distribution $p_0$ is a wide distribution, such as a Gaussian distribution with mean zero $\mu_0 = \mathbf{0}$ and a diagonal covariance $\mathbf{\Sigma}_0$ of large magnitude.

CEM attempts to find the single-best set of optimal parameters for a single optimization problem. The straightforward use in reinforcement learning is to learn the single-best set of policy parameters $\mathbf{w}$ (Szita & Lörincz, 2006; Mannor et al., 2003). Our goal, however, is to (repeatedly) find maximally valued actions $a^*$ conditioned on each state for $Q(s, \cdot)$. The global optimization strategy could be run on each step to find the exact best action for each current

---

**Algorithm 1** Percentile Empirical Distribution($N, \rho$)

Evaluate and sort in descending order:
$Q_\theta(S_t, a_{i_1}) \geq \ldots \geq Q_\theta(S_t, a_{i_N})$
**return** $\hat{I}(S_t) = \{a_{i_1}, \ldots, a_{i_h}\}$
(where $h = \lceil \rho N \rceil$ )

---

state, as in QT-Opt (Kalashnikov et al., 2018) and follows-ups (Simmons-Edler et al., 2019; Shao et al., 2022), but this is expensive and throws away prior information about the function surface obtained when previous optimizations were executed.

We extend CEM to be (a) conditioned on state and (b) learned iteratively over time. The key modification when extending CEM to Conditional CEM (CCEM), to handle these two key differences, is to introduce another *proposal policy* that concentrates more slowly. This proposal policy is entropy-regularized to ensure that we keep a broader set of potential actions when sampling, in case changing action-values are very different since the previous update to that state. The main policy (the actor) does not use entropy regularization, allowing it to more quickly start acting according to currently greedy actions, without collapsing. We visualize this in Figure 1.

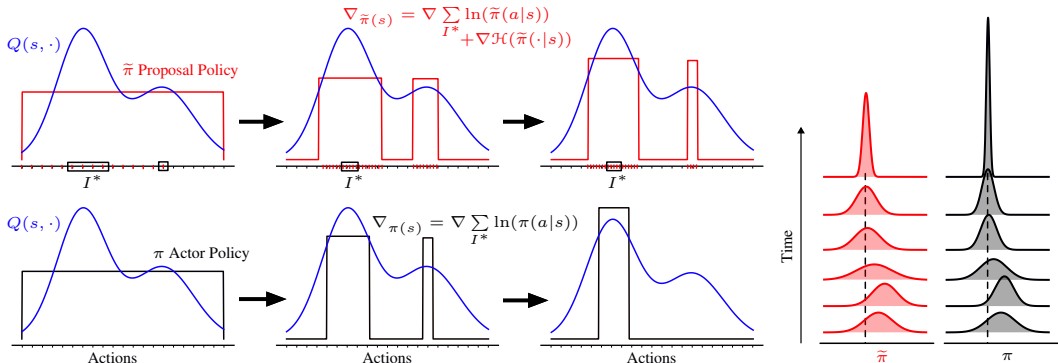

Figure 1: In the **left** figure we see multiple updates for both policies of the CCEM in a single state. We use uniform policies, for interpretability. In the **rightmost** figure, we show an actual progression of CCEM with Gaussian policies, when executed on the action-values depicted in the leftmost figure. The Actor policy (in black) concentrates more quickly than the Proposal policy (in red).

The CCEM algorithm is presented in Algorithm 2. On each step, the proposal policy, $\tilde{\pi}_{\mathbf{w}'_t}(\cdot|S_t)$, is sampled to provide a set of actions $a_1, \ldots, a_N$ from which we construct the empirical distribution $\hat{I}(S_t) = \{a_1^*, \ldots, a_h^*\}$ of maximally valued actions. The actor parameters $\mathbf{w}_t$ are updated using a gradient ascent step on the log-likelihood of the actions $\hat{I}(S_t)$. The proposal parameters $\mathbf{w}'_t$ are updated using a similar update, but with an entropy regularizer. To obtain $\hat{I}(S_t)$, we select $a_i^* \subset \{a_1, \ldots, a_N\}$ where $Q(S_t, a_i^*)$ are in the top $(1-\rho)$ quantile values. For example, for $\rho = 0.2$, approximately the top 20% of actions are chosen, with $h = \lceil \rho N \rceil$. Implicitly, $f_t$ is $Q_\theta(S_t, a_h^*)$ for $a_h^*$ the action with the lowest value in this top percentile. This procedure is summarized in Algorithm 1.

Greedy Actor-Critic, in Algorithm 3, puts this all together. We use experience replay, and the CCEM algorithm on a mini-batch. The updates involve obtaining the sets $\hat{I}(S)$ for every $S$ in the mini-batch $B$ and updating with the gradient $\frac{1}{|B|} \sum_{S \in B} \sum_{a \in \hat{I}(S)} \nabla_{\mathbf{w}} \ln \pi_{\mathbf{w}}(a|S)$. The Sarsa update to the critic involves (1) sampling an on-policy action from the actor $A' \sim \pi_{\mathbf{w}}(\cdot|S')$ for each tuple in the mini-batch and (2) using the update $\frac{1}{|B|} \sum_{(S,A,S',R,A') \in B}(R + \gamma Q_\theta(S', A') - Q_\theta(S, A))\nabla_\theta Q_\theta(S, A)$. Other critic updates are possible; we discuss alternatives and connections to related algorithms in Appendix A.

---

**Algorithm 2** Conditional CEM for the Actor

**Input:** $S_t$ and $Q_\theta$, $N \in \mathbb{N}$, $\rho \in (0,1)$
**if** actions discrete and $|\mathcal{A}| \leq 1/\rho$ **then**
    $\hat{I}(S_t) = \arg\max_{a \in \mathcal{A}} Q_\theta(S_t, a)$
**else**
    Sample $N$ actions $a_i \sim \tilde{\pi}_{\mathbf{w}'}(\cdot|S_t)$
    Obtain $\hat{I}(S_t)$ using Algorithm 1
**end if**
$\mathbf{w} \leftarrow \mathbf{w} + \alpha_{p,t} \sum_{a \in \hat{I}(S_t)} \nabla_{\mathbf{w}} \ln \pi_{\mathbf{w}}(a|S_t)$

$\mathbf{w}' \leftarrow \mathbf{w}' + \alpha_{p,t}[\sum_{a \in \hat{I}(S_t)} \nabla_{\mathbf{w}'} \ln \tilde{\pi}_{\mathbf{w}'}(a|S_t) + \tau \nabla_{\mathbf{w}'} \mathcal{H}(\tilde{\pi}_{\mathbf{w}'}(\cdot|S_t))]$

---

**Algorithm 3** Greedy Actor-Critic

Initialize parameters $\theta, \mathbf{w}, \mathbf{w}'$, replay buffer $\mathcal{B}$
Obtain initial state $S$
**while** agent interacting with the environment
**do**
    Take action $A \sim \pi_{\mathbf{w}}(\cdot|S)$, observe $R, S'$
    Add $(S, A, S', R)$ to the buffer $\mathcal{B}$
    Grab a random mini-batch $B$ from buffer $\mathcal{B}$
    Update $\theta$ using Sarsa for policy $\pi_{\mathbf{w}}$ on $B$
    Update $\mathbf{w}, \mathbf{w}'$ using Algorithm 2 on $B$.
**end while**

---

**CCEM for Discrete Actions.** Although we can use the same algorithm for discrete actions, we can make it simpler when we have a small number of discrete actions. Our algorithm is designed around

the fact that it is difficult to solve for the maximal action for $Q_\theta(S_t, a)$ for continuous actions; we slowly identify this maximal action across states. For a small set of discrete actions, it is easy to get this maximizing action. If $|\mathscr{A}| < 1/\rho$, then the top percentile consists of the one top action (or the top actions if there are ties); we can directly set $\hat{I}(S_t) = \arg\max_{a \in \mathscr{A}} Q_\theta(S_t, a)$ and do not need to maintain a proposal policy. For this reason, we focus our theory on the continuous-action setting, which is the main motivation for using CEM for the actor update.

## 4    THEORETICAL GUARANTEES

In this section, we motivate that the target policy underlying CCEM guarantees policy improvement, and characterize the ODE underlying CCEM. We show it tracks a CEM update in expectation across states and slowly concentrates around maximally valued actions even while the action-values are changing.

### 4.1    POLICY IMPROVEMENT UNDER AN IDEALIZED SETTING

We first consider the setting where we have access to $Q^\pi$, as is typically done for characterizing the policy improvement properties of an operator (Haarnoja et al., 2018a; Ghosh et al., 2020; Chan et al., 2021) as well as for the original policy improvement theorem (Sutton & Barto, 2018). Our update moves our policy towards a *percentile-greedy* policy that redistributes probability solely to the $(1 - \rho)$-quantile according to magnitudes under $Q(s, a)$. More formally, let $f_Q^\rho(\pi; s)$ be the threshold such that $\int_{\{a \in \mathscr{A}|Q(s,a) \geq f_Q^\rho(\pi;s)\}} \pi(a|s)da = \rho$, namely that gives the set of actions in the top $1 - \rho$ quantile, according to magnitudes under $Q(s, \cdot)$. Then we can define the percentile-greedy policy as

$$\pi_\rho(a|s, Q, \pi) = \begin{cases} \pi(a|s)/\rho & Q(s, a) \geq \text{thresh} f_Q^\rho(\pi; s) \\ 0 & \text{else} \end{cases} \tag{2}$$

where diving by $\rho$ renormalizes the distribution. Computing this policy would be onerous; instead, we only sample the KL divergence to this policy, using a sample percentile. Nonetheless, this percentile-greedy policy represents the target policy that the actor updates towards (in the limit of samples $N$ for the percentile).

Intuitively, this target policy should give policy improvement, as it redistributes weight for low valued actions proportionally to high-valued actions. We formalize this in the following theorem. We write $\pi_\rho(a|s)$ instead of $\pi_\rho(a|s, Q^\pi, \pi)$, when it is clear from context.

**Theorem 4.1.** *For a given policy $\pi$, action-value $Q^\pi$ and $\rho > 0$, the percentile-greedy policy $\pi_\rho$ in $\pi$ and $Q^\pi$ is guaranteed to be at least as good as $\pi$ in all states:*

$$\int_{\mathscr{A}} \pi_\rho(a|s, Q^\pi, \pi)Q^{\pi_\rho}(s, a)da \geq \int_{\mathscr{A}} \pi(a|s)Q^\pi(s, a)da$$

*Proof.* The proof is a straightforward modification of the policy improvement theorem. Notice that

$$\int_{\mathscr{A}} \pi_\rho(a|s)Q^\pi(s, a)da \quad = \quad \int_{\{a \in \mathscr{A}|Q(s,a) \geq f_Q^\rho(\pi;s)\}} \frac{\pi(a|s)}{\rho} Q^\pi(s, a)da \quad \geq \quad \int_{\mathscr{A}} \pi(a|s)Q^\pi(s, a)da$$

by the definition of percentiles, for any state $s$. Rewriting $\int_{\mathscr{A}} \pi(a|s)Q^\pi(s, a)da = \mathbb{E}_\pi[Q^\pi(s, A)]$,

$$\begin{aligned} V^\pi(s) = \mathbb{E}_\pi[Q^\pi(s, A)] &\leq \mathbb{E}_{\pi_\rho}[Q^\pi(s, A)] = \mathbb{E}_{\pi_\rho}[R_{t+1} + \gamma\mathbb{E}_\pi[Q^\pi(S_{t+1}, A_{t+1})|S_t = s] \\ &\leq \mathbb{E}_{\pi_\rho}[R_{t+1} + \gamma\mathbb{E}_{\pi_\rho}[Q^\pi(S_{t+1}, A_{t+1})]|S_t = s] \\ &\leq \mathbb{E}_{\pi_\rho}[R_{t+1} + \gamma R_{t+2} + \gamma^2\mathbb{E}_\pi[Q^\pi(S_{t+2}, A_{t+2})]|S_t = s] \\ &\cdots \\ &\leq \mathbb{E}_{\pi_\rho}[R_{t+1} + \gamma R_{t+2} + \gamma^2 R_{t+3} + \ldots\gamma^{T-1}R_T|S_t = s] = \mathbb{E}_{\pi_\rho}[Q^{\pi_\rho}(s, A)] = V^{\pi_\rho}(s) \; \square \end{aligned}$$

This result is a sanity check to ensure the target policy is sensible in our update. Note that the Boltzmann policy only guarantees improvement under the entropy-regularized action-values.

## 4.2 CCEM TRACKS THE GREEDY ACTION

Beyond the idealized setting, we would like to understand the properties of the stochastic algorithm. CCEM is not a gradient descent approach, so we need to reason about its dynamics—namely the underlying ODE. We expect CCEM to behave like CEM per state, but with some qualifiers. First, CCEM uses a parameterized policy conditioned on state, meaning that there is aliasing between the action distributions per state. CEM, on the other hand, does not account for such aliasing. We identify conditions on the parameterized policy and use an ODE that takes expectations over states.

Second, the function we are attempting to maximize is also changing with time, because the action-values are updating. We address this issue using a two-timescale stochastic approximation approach, where the action-values $Q_\theta$ change more slowly than the policy, allowing the policy to track the maximally valued actions. The policy itself has two timescales, to account for its own parameters changing at different timescales. Actions for the maximum likelihood step are selected according to older (slower) parameters $\mathbf{w}'$, so that it is as if the primary (faster) parameters $\mathbf{w}$ are updated using samples from a fixed distribution. These two policies correspond to our proposal policy (slow) and actor (fast).

We show that the ODE for the CCEM parameters $\mathbf{w}_t$ is based on the gradient

$$\nabla_{\mathbf{w}(t)} \mathbb{E}_{S \sim \nu, A \sim \pi_{\mathbf{w}'}(\cdot|S)} \left[ I_{\{Q_\theta(S,A) \geq f_\theta^\rho(\mathbf{w}';S)\}} \ln \pi_{\mathbf{w}(t)}(A|S) \right]$$

where $\theta$ and $\mathbf{w}'$ are changing at slower timescales, and so effectively fixed from the perspective of the faster changing $\mathbf{w}_t$. The term per-state is exactly the update underlying CEM, and so we can think of this ODE as one for an expected CEM Optimizer, across states for parameterized policies. We say that CCEM *tracks* this expected CEM Optimizer, because $\theta$ and $\mathbf{w}'$ are changing with time.

We provide an informal theorem statement here for Theorem B.1, with a proof-sketch. The main result, including all conditions, is given in Appendix B. We discuss some of the (limitations of the) conditions after the proof sketch.

**Informal Result:** Let $\theta_t$ be the action-value parameters with stepsize $\alpha_{q,t}$, and $\mathbf{w}_t$ be the policy parameters with stepsize $\alpha_{a,t}$, with $\mathbf{w}'_t$ a more slowly changing set of policy parameters set to $\mathbf{w}'_t = (1 - \alpha'_{a,t})\mathbf{w}'_t + \alpha'_{a,t}\mathbf{w}_t$ for stepsize $\alpha'_{a,t} \in (0,1]$. Assume: (1) States $S_t$ are sampled from a fixed marginal distribution. (2) $\nabla_{\mathbf{w}} \ln \pi_{\mathbf{w}}(\cdot|s)$ is locally Lipschitz w.r.t. $\mathbf{w}$, $\forall s \in \mathcal{S}$. (3) Parameters $\mathbf{w}_t$ and $\theta_t$ remain bounded almost surely. (4) Stepsizes are chosen for three different timescales: $\mathbf{w}_t$ evolves faster than $\mathbf{w}'_t$ and $\mathbf{w}'_t$ evolves faster than $\theta_t$. Under these four conditions, the CCEM Actor tracks the expected CEM Optimizer.

**Proof Sketch:** The stochastic update to the Actor is not a direct gradient-descent update. Each update to the Actor is a CEM update, which requires a different analysis to ensure that the stochastic noise remains bounded and is asymptotically negligible. Further, the classical results of CEM also do not immediately apply, because such updates assume distribution parameters can be directly computed. Here, distribution parameters are conditioned on state, as outputs from a parametrized function. We identify conditions on the parametrized policy to ensure well-behaved CEM updates.

The multi-timescale analysis allows us to focus on the updates of the Actor $\mathbf{w}_t$, assuming the action-value parameter $\theta$ and action-sampling parameter $\mathbf{w}'$ are quasi-static. These parameters are allowed to change with time—as they will in practice—but are moving at a sufficiently slower timescale relative to $\mathbf{w}_t$ and hence the analysis can be undertaken as if they are static.

The first step in the proof is to formulate the update to the weights as a projected stochastic recursion—simply meaning a stochastic update where after each update the weights are projected to a compact, convex set to keep them bounded. The stochastic recursion is reformulated into a summation involving the mean vector field $g^\theta(\mathbf{w}_t)$ (which depends on the action-value parameters $\theta$), martingale noise, and a loss term $\ell_t^\theta$ that is due to having approximate quantiles. The key steps are then to show almost surely that the mean vector field $g^\theta$ is locally Lipschitz, the martingale noise is quadratically bounded and that the loss term $\ell_t^\theta$ decays to zero asymptotically. For the first and second, we identify conditions on the policy parameterization that guarantee these. For the final case, we adapt the proof for sampled quantiles approaching true quantiles for CEM, with modifications to account for expectations over the conditioning variable, the state. ∎

This result has several limitations. First, it does not perfectly characterize the CCEM algorithm that we actually use. We do not use the update $\mathbf{w}'_t = (1 - \alpha'_{a,t})\mathbf{w}'_t + \alpha'_{a,t}\mathbf{w}_t$, and instead use entropy

regularization to make $\mathbf{w}'_t$ concentrate more slowly than $\mathbf{w}_t$. The principle is similar; empirically we found entropy regularization to be an effective strategy to achieve this condition.

Second, the theory assumes the state distribution is fixed, and not influenced by $\pi_{\mathbf{w}}$. It is standard to analyze the properties of (off-policy) algorithms for fixed datasets as a first step, as was done for Q-learning (Jaakkola et al., 1994). It allows us to separate concerns, and just ask: does our method concentrate on maximal actions across states? An important next step is to characterize the full Greedy Actor-Critic algorithm, beyond just understanding the properties of the CCEM component.

## 5 EMPIRICAL RESULTS

We are primarily interested in investigating sensitivity to hyperparameters. This sensitivity reflects how difficult it can be to get AC methods working on a new task—relevant for both applied settings and research. AC methods have been notoriously difficult to tune due to the interacting time scales of the actor and critic (Degris et al., 2012a), further compounded by the sensitivity in the entropy scale. The use of modern optimizers may reduce some of the sensitivity in stepsize selection; these experiments help understand if that is the case. Further, a very well-tuned algorithm may not be representative of performance across problems. We particularly examine the impacts of selecting a single set of hyperparameters across environments, in contrast to tuning per environment.

We chose to conduct experiments in small, challenging domains appropriately sized for extensive experiment repetition. Ensuring significance in results and carefully exploring hyperparameter sensitivity required many experiments. Our final plots required $\sim$30,000 runs across all environments, algorithms, and hyperparameters. Further, contrary to popular belief, classic control domains are a challenge for Deep RL agents (Ghiassian et al., 2020), and performance differences in these environments have been shown to extend to larger environments (Obando-Ceron & Castro, 2021).

### 5.1 ALGORITHMS

We focus on comparing GreedyAC to Soft Actor-Critic (SAC) both since this allows us to compare to a method that uses the Boltzmann target policy on action-values and because SAC continues to have the most widely reported success[2]. We additionally include VanillaAC as a baseline, a basic AC variant which does not include any of the tricks SAC utilizes to improve performance, such as action reparameterization to estimate the policy gradient or double Q functions to mitigate maximization bias. For discrete actions, policies are parameterized using Softmax distributions. For continuous actions, policies are parameterized using Gaussian distributions, except SAC which uses a squashed Gaussian policy as per the original work. We tested SAC with a Gaussian policy, and it performed worse. All algorithms use neural networks. Feedforward networks consist of two hidden layers of 64 units (classic control environments) or 256 units (Swimmer-v3 environment). Convolutional layers consists of one convolutional layer with 3 kernels of size 16 followed by a fully connected layer of size 128. All algorithms use the Adam optimizer (Kingma & Ba, 2014), experience replay, and target networks for the value functions. See Appendix C.1 for a full discussion of hyperparameters.

### 5.2 ENVIRONMENTS

We use the classic versions of Mountain Car (Sutton & Barto, 2018), Pendulum (Degris et al., 2012a), and Acrobot (Sutton & Barto, 2018). Each environment is run with both continuous and discrete action spaces; states are continuous. Discrete actions consist of the two extreme continuous actions and 0. All environments use a discount factor of $\gamma = 0.99$, and episodes are cut off at 1,000 timesteps, teleporting the agent back to the start state (but not causing termination). To demonstrate the potential of GreedyAC at scale, we also include experiments on Freeway and Breakout from MinAtar (Young & Tian, 2019) as well as on Swimmer-v3 from OpenAI Gym (Brockman et al., 2016). On MinAtar, episodes are cutoff at 2,500 timesteps.

In Mountain Car, the goal is to drive an underpowered car up a hill. State consists of the position in $[-1.2, 0.6]$ and velocity in $[-0.7, 0.7]$. The agent starts in a random position in $[-0.6, -0.4]$ and velocity 0. The action is the force to apply to the car, in $[-1, 1]$. The reward is -1 per step.

In Pendulum, the goal is to hold a pendulum with a fixed base in a vertical position. State consists of the angle (normalized in $[-\pi, \pi)$) and angular momentum (in $[-1, 1]$). The agent starts with the pendulum facing downwards and 0 velocity. The action is the torque applied to the fixed base, in $[-2, 2]$. The reward is the cosine of the angle of the pendulum from the positive y-axis.

---

[2]See https://spinningup.openai.com/en/latest/spinningup/bench.html

In Acrobot, the agent controls a doubly-linked pendulum with a fixed base. The goal is to swing the second link one link's length above the fixed base. State consists of the angle of each link (in $[-\pi, \pi)$) and the angular velocity of each link (in $[-4\pi, 4\pi]$ and $[-9\pi, 9\pi]$ respectively). The agent starts with random angles and angular velocities in $[-0.1, 0.1]$. The action is the torque applied to the joint between the two links, in $[-1, 1]$. The reward is -1 per step.

### 5.3 EXPERIMENTAL DETAILS

We sweep hyperparameters for 40 runs, tuning over the first 10 runs and reporting results using the final 30 runs for the best hyperparameters. We sweep critic step size $\alpha = 10^x$ for $x \in \{-5, -4, \ldots, -1\}$. We set the actor step size to be $\kappa \times \alpha$ and sweep $\kappa \in \{10^{-3}, 10^{-2}, 10^{-1}, 1, 2, 10\}$. We sweep entropy scales $\tau = 10^y$ for $y \in \{-3, -2, -1, 0, 1\}$. For the classic control experiments, we used fixed batch sizes of 32 samples and a replay buffer capacity of 100,000 samples. For the MinAtar experiments, we used fixed batch sizes of 32 samples and a buffer capacity of 1 million. For the Swimmer experiments, we used fixed batch sizes of 100 samples and a buffer capacity of 1 million. For CCEM, we fixed $\rho = 0.1$ and sample $N = 30$ actions.

To select hyperparameters across environment, we must normalize performance to provide an aggregate score. We use near-optimal performance as the normalizer for each environment, with a score of 1 meaning equal to this performance. We only use this normalization to average scores across environments. We report learning curves using the original unnormalized returns. For more details, see Appendix C.2.

### 5.4 RESULTS

**Per-environment Tuning:** We first examine how well the algorithms can perform when they are tuned per-environment. In Figure 2, we see that SAC performs well in Pendulum-CA (continuous actions) and in Pendulum-DA (discrete actions) but poorly in the other settings. SAC learns slower than GreedyAC and VanillaAC on Acrobot. GreedyAC performs worse than SAC in Pendulum-CA, but still performs acceptably, nearly reaching the same final performance. SAC performs poorly on both versions of Mountain

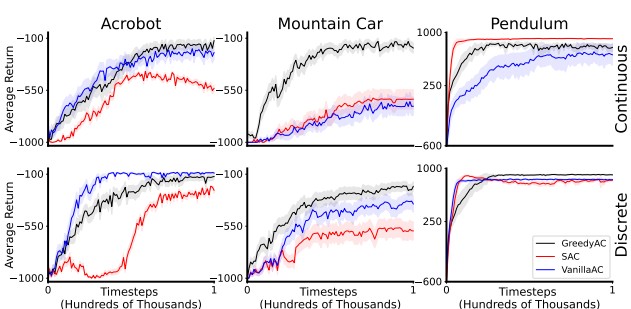

Figure 2: Learning curves when tuning hyperparameters **per-environment**, averaged over 30 runs with standard errors.

Car. That AC methods struggle with Acrobot is common wisdom, but here we see that both GreedyAC and VanillaAC do well on this problem. GreedyAC is the clear winner in Mountain Car.

**Across-environment Tuning:** We next examine the performance of the algorithms when they are forced to select one hyperparameter setting across continuous- or discrete-action environments separately, shown in Figure 3. We expect algorithms that are less sensitive to their parameters to suffer less degradation. Under this regime, GreedyAC has a clear advantage over SAC. GreedyAC maintains acceptable performance across all environments, sometimes learning more slowly than under per-environment

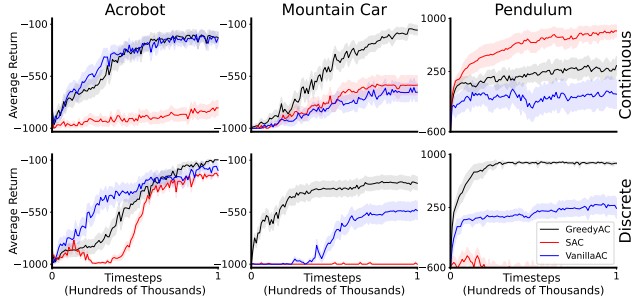

Figure 3: Learning curves when tuning hyperparameters **across-environments**, averaged over 30 runs with standard errors.

tuning, but having reasonable behavior. SAC performs poorly on two-thirds of the environments. GreedyAC is less sensitive than VanillaAC under across-environment tuning and performs at least as good as VanillaAC.

**Hyperparameter Sensitivity:** We examine the sensitivity of GreedyAC and SAC to their entropy scales, focusing on the continuous action environments. We plot sensitivity curves, with one plotted for each entropy scale, with the stepsize on the x-axis and average return across all steps and all 40 runs on the y-axis. Because there are two stepsizes, we have two sets of plots. When examining the sensitivity to the critic stepsize, we select the best actor stepsize. We do the same for the actor stepsize plots. We provide the plots with individual lines in Appendix C.3 and here focus on a more summarized view.

Figure 4 depicts the range of performance obtained across entropy scales. The plot is generated by filling in the region between the curves for each entropy scale. If this *sensitivity region* is broad, then the algorithm performed very differently across different entropy scales and so is sensitive to the entropy. SAC has much wider sensitivity regions than GreedyAC. Those of GreedyAC are generally narrow, indicating that the stepsize rather than entropy was the dominant factor. Further, the bands of performance are

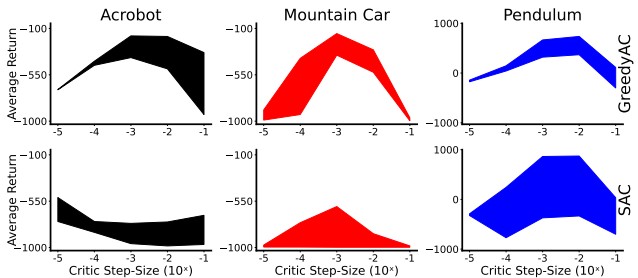

Figure 4: A **sensitivity region** plot for entropy, for GreedyAC (top row) and SAC (bottom row) in the continuous action problems.

generally at the top of the plot. When SAC exhibits narrower regions than GreedyAC, those regions are lower on the plot, indicating overall poor performance.

## 6 SCALING GREEDY-AC

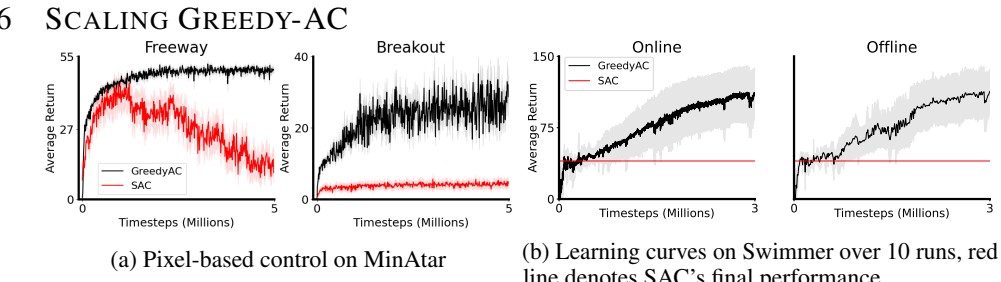

(a) Pixel-based control on MinAtar

(b) Learning curves on Swimmer over 10 runs, red line denotes SAC's final performance

Figure 5: Comparing GreedyAC and SAC in more challenging environments

We ran GreedyAC on two pixel-based control problems, Breakout and Freeway, from the MinAtar suite (Young & Tian, 2019). Recent work has shown that MinAtar results are indicative of those in much larger scale problems (Obando-Ceron & Castro, 2021). We set the actor step-size scale to 1.0 and a critic step-size of $10^{-3}$ for both GreedyAC and SAC—the defaults of SAC. We set the entropy scale of SAC to $10^{-3}$ based on a grid search. Figure 5a above clearly indicates GreedyAC can learn a good policy from high-dimensional inputs; comparable performance to DQN Rainbow.

Finally, we ran GreedyAC on Swimmer-v3 from OpenAI Gym (Brockman et al., 2016). We tuned over one run and then ran the tuned hyperparameters for an additional 9 runs to generate Figure 5b. We report online and offline performance. Offline evaluation is performed every 10,000 steps, for 10 episodes, where only the mean action is selected and learning is disabled. We report SAC's final performance on Swimmer from the SpinningUp benchmark[3]. GreedyAC is clearly not state-of-the-art here—most methods are not—however, GreedyAC steadily improves throughout the experiment.

## 7 CONCLUSION

In this work, we introduced a new Actor-Critic (AC) algorithm called GreedyAC, that uses a new update to the Actor based on an extension of the cross-entropy method (CEM). The idea is to (a) define a *percentile-greedy* target policy and (b) update the actor towards this target policy, by reducing a KL divergence to it. This percentile-greedy policy guarantees policy improvement, and we prove that our Conditional CEM algorithm tracks the actions of maximal value under changing action-values. We conclude with an in-depth empirical study, showing that GreedyAC has significantly lower sensitivity to its hyperparameters than SAC does.

---

[3]See https://spinningup.openai.com/en/latest/spinningup/bench.html

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

## A    RELATED POLICY OPTIMIZATION ALGORITHMS

As mentioned in the main text, there are many policy optimization algorithms that can be seen as approximate policy iteration (API) rather than performing gradient descent on a policy objective. An overview and survey are given by Vieillard et al. (2020a) and Chan et al. (2021). There, many methods are shown to either minimize a forward or reverse KL-divergence to the Boltzmann policy. Our approach similarly updates the actor using a KL-divergence to a target policy, but here that target policy is the percentile-greedy policy. By doing a maximum likelihood update with actions sampled under the percentile-greedy policy, we are reducing the forward KL-divergence to the percentile-greedy policy in Equation 2.

Our CCEM update for the actor is new, but there are several approaches that resemble the idea, particularly those that try to match an expert. This includes dual policy iteration methods (DPI) (Parisotto et al., 2016; Sun et al., 2018; Steckelmacher et al., 2019) and RL as classification methods (Lagoudakis & Parr, 2003; Lazaric et al., 2010; Farahmand et al., 2015). DPI has two policies, one which is guiding the other. For example, one policy might be an expensive tree search and another a learned neural network, trained to mimic the first (expert or guide) policy. CCEM, on the other hand, uses two policies differently. Our actor does not imitate our proposal policy. Rather, the proposal policy is used to improve the search over the nonconcave surface of Q. It samples actions more broadly, to make it more likely to find a maximizing action. Further, the actor increases the likelihood of only the top actions and does not imitate the proposal policy. In contrast, Bootstrap DPI (Steckelmacher et al., 2019, Equation 5) uses an update based on Actor-Mimic (Parisotto et al., 2016), where the policy increases likelihood of actions for the softmax policies it is trying to mimic. The

resemblance arises from the fact that (Steckelmacher et al., 2019, Equation 5) can be seen as a sum over forward KL divergences to softmax policies (for discrete actions), just like we have a forward KL divergence but to the percentile-greedy policy (for discrete or continuous actions).

The other class of algorithms, RL as classification, also look similar due to using a forward KL divergence. They reduce the problem to identifying "positive" actions in a state (producing maximal returns) and "negative" actions in a state (producing non-maximal returns). If a cross-entropy loss is used, then this corresponds to maximizing the likelihood of the positive actions and minimizing the likelihood of the negative ones. More generally, other classification algorithms can be used that do not involve maximizing likelihood (like SVMs). The RL as classifications algorithms primarily focus on how to obtain these positive and negative actions, and otherwise look quite different from Greedy AC, in addition to being restricted to a discrete set of actions.

Finally, we can also consider the connection to Conservative Policy Iteration (CPI) (Kakade & Langford, 2002) and a generalization called Deep CPI (Vieillard et al., 2020b). CPI updates the policy to be an interpolation between the greedy policy $G(Q)$ and the current policy $\pi$, to get the new policy $\pi' = (1-\alpha)\pi + \alpha G(Q)$ for $\alpha \in [0, 1]$. Deep CPI extends this idea to parameterized policies, instead minimizing a forward KL to this interpolation policy. Greedy AC could be seen as another way to obtain a conservative update, because it does not move the actor all the way to the greedy policy. Instead, it moves towards the percentile-greedy policy (in Equation 2), which shifts probability to the upper percentile of actions. Similarly to the interpolation policy, this percentile-greedy policy depends on the previous policy and on increasing probability for maximally valued actions. As yet, Deep CPI has not been shown to enjoy the same theoretical guarantees as CPI: minimizing a forward KL to the interpolation policy does not provide the same guarantees. It remains an open question how to implement this conservative update in deep RL, and it would be interesting to understand if the CCEM update could provide an alternative route to obtaining such guarantees.

## B    CONVERGENCE ANALYSIS OF THE ACTOR

We provided an informal proof statement and proof sketch in Section 4.2, to provide intuition for the result. Here, we provide the formal proof in the following subsections. We first provide some definitions, particularly for the quantile function which is central to the analysis. We then lay out the assumptions, and discuss some policy parameterizations to satisfy those assumptions. We finally state the theorem, with proof, and provide one lemma needed to prove the theorem in the final subsection.

### B.1    NOTATION AND DEFINITIONS

**Notation:**    For a set $A$, let $\mathring{A}$ represent the interior of $A$, while $\partial A$ is the boundary of $A$. The abbreviation $a.s.$ stands for *almost surely* and $i.o.$ stands for *infinitely often*. Let $\mathbb{N}$ represent the set $\{0, 1, 2, \dots\}$. For a set $A$, we let $I_A$ to be the indicator function/characteristic function of $A$ and is defined as $I_A(x) = 1$ if $x \in A$ and $0$ otherwise. Let $\mathbb{E}_g[\cdot]$, $\mathbb{V}_g[\cdot]$ and $\mathbb{P}_g(\cdot)$ denote the expectation, variance and probability measure *w.r.t.* $g$. For a $\sigma$-field $\mathscr{F}$, let $\mathbb{E}[\cdot|\mathscr{F}]$ represent the conditional expectation *w.r.t.* $\mathscr{F}$. A function $f : X \to Y$ is called Lipschitz continuous if $\exists L \in (0, \infty)$ *s.t.* $\|f(\mathbf{x}_1) - f(\mathbf{x}_2)\| \leq L\|\mathbf{x}_1 - \mathbf{x}_2\|$, $\forall \mathbf{x}_1, \mathbf{x}_2 \in X$. A function $f$ is called locally Lipschitz continuous if for every $\mathbf{x} \in X$, there exists a neighbourhood $U$ of $X$ such that $f_{|U}$ is Lipschitz continuous. Let $C(X, Y)$ represent the space of continuous functions from $X$ to $Y$. Also, let $B_r(\mathbf{x})$ represent an open ball of radius $r$ with centered at $\mathbf{x}$. For a positive integer $M$, let $[M] \doteq \{1, 2 \dots M\}$.

**Definition 1.** *A function* $\Gamma : U \subseteq \mathbb{R}^{d_1} \to V \subseteq \mathbb{R}^{d_2}$ *is Frechet differentiable at* $\mathbf{x} \in U$ *if there exists a bounded linear operator* $\widehat{\Gamma}_{\mathbf{x}} : \mathbb{R}^{d_1} \to \mathbb{R}^{d_2}$ *such that the limit*

$$\lim_{\epsilon \downarrow 0} \frac{\Gamma(\mathbf{x} + \epsilon \mathbf{y}) - \mathbf{x}}{\epsilon} \tag{3}$$

*exists and is equal to* $\widehat{\Gamma}_{\mathbf{x}}(\mathbf{y})$. *We say* $\Gamma$ *is Frechet differentiable if Frechet derivative of* $\Gamma$ *exists at every point in its domain.*

**Definition 2.** *Given a bounded real-valued continuous function* $H : \mathbb{R}^d \to \mathbb{R}$ *with* $H(a) \in [H_l, H_u]$ *and a scalar* $\rho \in [0, 1]$, *we define the* $(1-\rho)$-*quantile of* $H(A)$ *w.r.t. the PDF* $g$ *(denoted as* $f^\rho(H, g)$*)*

*as follows:*

$$f^\rho(H, g) \doteq \sup_{\ell \in [H_l, H_u]} \{\mathbb{P}_g\big(H(A) \geq \ell\big) \geq \rho\}, \tag{4}$$

*where $\mathbb{P}_g$ is the probability measure induced by the PDF $g$, i.e., for a Borel set $\mathcal{A}$, $\mathbb{P}_g(\mathcal{A}) \doteq \int_{\mathcal{A}} g(a)da$.*

This quantile operator will be used to succinctly write the quantile for $Q_\theta(S, \cdot)$, with actions selected according to $\pi_{\mathbf{w}}$, i.e.,

$$f_\theta^\rho(\mathbf{w}; s) \doteq f^\rho(Q_\theta(s, \cdot), \pi_{\mathbf{w}}(\cdot|s)) = \sup_{\ell \in [Q_l^\theta, Q_u^\theta]} \{\mathbb{P}_{\pi_{\mathbf{w}}(\cdot|s)}\big(Q_\theta(s, A) \geq \ell\big) \geq \rho\}. \tag{5}$$

## B.2 ASSUMPTIONS

**Assumption 1.** *Given a realization of the transition dynamics of the MDP in the form of a sequence of transition tuples $\mathcal{O} \doteq \{(S_t, A_t, R_t, S_t')\}_{t \in \mathbb{N}}$, where the state $S_t \in \mathcal{S}$ is drawn using a latent sampling distribution $\nu$, while $A_t \in \mathcal{A}$ is the action chosen at state $S_t$, the transitioned state $S \ni S_t' \sim P(S_t, A_t, \cdot)$ and the reward $\mathbb{R} \ni R_t \doteq R(S_t, A_t, S_t')$. We further assume that the reward is uniformly bounded, i.e., $|R(\cdot, \cdot, \cdot)| < R_{max} < \infty$.*

We analyze the long run behaviour of the conditional cross-entropy recursion (actor) which is defined as follows:

$$\mathbf{w}_{t+1} \doteq \Gamma^W \left\{ \mathbf{w}_t + \alpha_{a,t} \frac{1}{N_t} \sum_{A \in \Xi_t} I_{\{Q_{\theta_t}(S_t, A) \geq \hat{f}_{t+1}^\rho\}} \nabla_{\mathbf{w}_t} \ln \pi_{\mathbf{w}}(A|S_t) \right\}, \tag{6}$$

$$\text{where } \Xi_t \doteq \{A_{t,1}, A_{t,2}, \ldots, A_{t,N_t}\} \stackrel{\text{iid}}{\sim} \pi_{\mathbf{w}_t'}(\cdot|S_t).$$

$$\mathbf{w}_{t+1}' \doteq \mathbf{w}_t' + \alpha_{a,t}' \left(\mathbf{w}_{t+1} - \mathbf{w}_t'\right). \tag{7}$$

Here, $\Gamma^W\{\cdot\}$ is the projection operator onto the compact (closed and bounded) and convex set $W \subset \mathbb{R}^m$ with a smooth boundary $\partial W$. Therefore, $\Gamma^W$ maps vectors in $\mathbb{R}^m$ to the nearest vectors in $W$ w.r.t. the Euclidean distance (or equivalent metric). Convexity and compactness ensure that the projection is unique and belongs to $W$.

**Assumption 2.** *The pre-determined, deterministic, step-size sequences $\{\alpha_{a,t}\}_{t \in \mathbb{N}}$, $\{\alpha_{a,t}'\}_{t \in \mathbb{N}}$ and $\{\alpha_{q,t}\}_{t \in \mathbb{N}}$ are positive scalars which satisfy the following:*

$$\sum_{t \in \mathbb{N}} \alpha_{a,t} = \sum_{t \in \mathbb{N}} \alpha_{a,t}' = \sum_{t \in \mathbb{N}} \alpha_{q,t} = \infty$$

$$\sum_{t \in \mathbb{N}} \left(\alpha_{a,t}^2 + \alpha_{a,t}'^2 + \alpha_{q,t}^2\right) < \infty$$

$$\lim_{t \to \infty} \frac{\alpha_{a,t}'}{\alpha_{a,t}} = 0, \quad \lim_{t \to \infty} \frac{\alpha_{q,t}}{\alpha_{a,t}} = 0.$$

The first conditions in Assumption 2 are the classical Robbins-Monro conditions (Robbins & Monro, 1985) required for stochastic approximation algorithms. The last two conditions enable the different stochastic recursions to have separate timescales. Indeed, it ensures the $\mathbf{w}_t$ recursion is faster compared to the recursions of $\theta_t$ and $\mathbf{w}_t'$. This timescale divide is needed to obtain the desired asymptotic behaviour, as we describe in the next section.

**Assumption 3.** *The pre-determined, deterministic, sample length schedule $\{N_t \in \mathbb{N}\}_{t \in \mathbb{N}}$ is positive and strictly monotonically increases to $\infty$ and $\inf_{t \in \mathbb{N}} \frac{N_{t+1}}{N_t} > 1$.*

Assumption 3 states that the number of samples increases to infinity and is primarily required to ensure that the estimation error arising due to the estimation of sample quantiles eventually decays to 0. Practically, one can indeed consider a fixed, finite, positive integer for $N_t$ which is large enough to accommodate the acceptable error.

**Assumption 4.** *The sequence $\{\theta_t\}_{t\in\mathbb{N}}$ satisfies $\theta_t \in \Theta$, where $\Theta \subset \mathbb{R}^n$ is a convex, compact set. Also, for $\theta \in \Theta$, let $Q_\theta(s,a) \in [Q_l^\theta, Q_u^\theta]$, $\forall s \in \mathcal{S}, a \in \mathcal{A}$.*

Assumption 4 assumes stability of the Expert, and minimally only requires that the values remain in a bounded range. We make no additional assumptions on the convergence properties of the Expert, as we simply need stability to prove the Actor tracks the update.

**Assumption 5.** *For $\theta \in \Theta$ and $s \in \mathcal{S}$, let $\mathbb{P}_{A\sim\pi_{\mathbf{w}'}(\cdot|s)}\left(Q_\theta(s,A) \geq \ell\right) > 0$, $\forall \ell \in [Q_l^\theta, Q_u^\theta]$ and $\forall \mathbf{w}' \in W$.*

Assumption 5 implies that there always exists a strictly positive probability mass beyond every threshold $\ell \in [Q_l^\theta, Q_u^\theta]$. This assumption is easily satisfied when $Q_\theta(s,a)$ is continuous in $a$ and $\pi_{\mathbf{w}}(\cdot|s)$ is a continuous probability density function.

**Assumption 6.**

$$\sup_{\substack{\mathbf{w},\mathbf{w}'\in W, \\ \theta\in\Theta, \ell\in[Q_l^\theta,Q_u^\theta]}} \mathbb{E}_{A\sim\pi_{\mathbf{w}'}(\cdot|S)}\left[\left\|I_{\{Q_\theta(S,A)\geq\ell\}}\nabla_{\mathbf{w}}\ln\pi_{\mathbf{w}}(A|S) - \mathbb{E}_{A\sim\pi_{\mathbf{w}'}(\cdot|S)}\left[I_{\{Q_\theta(S,A)\geq\ell\}}\nabla_{\mathbf{w}}\ln\pi_{\mathbf{w}}(A|S)\big|S\right]\right\|_2^2\bigg|S\right] < \infty \quad a.s.,$$

$$\sup_{\substack{\mathbf{w},\mathbf{w}'\in W, \\ \theta\in\Theta, \ell\in[Q_l^\theta,Q_u^\theta]}} \mathbb{E}_{A\sim\pi_{\mathbf{w}'}(\cdot|S)}\left[\left\|I_{\{Q_\theta(S,A)\geq\ell\}}\nabla_{\mathbf{w}}\ln\pi_{\mathbf{w}}(A|S)\right\|_2^2\bigg|S\right] < \infty \quad a.s.$$

**Assumption 7.** *For $s \in \mathcal{S}$, $\nabla_{\mathbf{w}}\ln\pi_{\mathbf{w}}(\cdot|s)$ is locally Lipschitz continuous w.r.t. $\mathbf{w}$.*

Assumptions 6 and 7 are technical requirements that can be more easily characterized when we consider $\pi_{\mathbf{w}}$ to belong to the natural exponential family (NEF) of distributions.

**Definition 3.** *Natural exponential family of distributions (NEF)(Morris, 1982): These probability distributions over $\mathbb{R}^m$ are represented by*

$$\{\pi_\eta(\mathbf{x}) \doteq h(\mathbf{x})e^{\eta^\top T(\mathbf{x})-K(\eta)} \mid \eta \in \Lambda \subset \mathbb{R}^d\}, \tag{8}$$

*where $\eta$ is the natural parameter, $h : \mathbb{R}^m \longrightarrow \mathbb{R}$, while $T : \mathbb{R}^m \longrightarrow \mathbb{R}^d$ (called the sufficient statistic) and $K(\eta) \doteq \ln\int h(\mathbf{x})e^{\eta^\top T(\mathbf{x})}d\mathbf{x}$ (called the cumulant function of the family). The space $\Lambda$ is defined as $\Lambda \doteq \{\eta \in \mathbb{R}^d \mid |K(\eta)| < \infty\}$. Also, the above representation is assumed minimal.[4] A few popular distributions which belong to the NEF family include Binomial, Poisson, Bernoulli, Gaussian, Geometric and Exponential distributions.*

We parametrize the policy $\pi_{\mathbf{w}}(\cdot|S)$ using a neural network, which implies that when we consider NEF for the stochastic policy, the natural parameter $\eta$ of the NEF is being parametrized by $\mathbf{w}$. To be more specific, we have $\{\psi_{\mathbf{w}} : \mathcal{S} \to \Lambda | \mathbf{w} \in \mathbb{R}^m\}$ to be the function space induced by the neural network of the actor, *i.e.*, for a given state $s \in \mathcal{S}$, $\psi_{\mathbf{w}}(s)$ represents the natural parameter of the NEF policy $\pi_{\mathbf{w}}(\cdot|s)$. Further,

$$\begin{aligned}\nabla_{\mathbf{w}}\ln\pi_{\mathbf{w}}(A|S) &= \ln(h(A)) + \psi_{\mathbf{w}}(S_t)^\top T(A) - K(\psi_{\mathbf{w}}(S)) \\ &= \nabla_{\mathbf{w}}\psi_{\mathbf{w}}(S)\left(T(A) - \nabla_\eta K(\psi_{\mathbf{w}}(S))\right). \\ &= \nabla_{\mathbf{w}}\psi_{\mathbf{w}}(S)\left(T(A) - \mathbb{E}_{A\sim\pi_{\mathbf{w}}(\cdot|S)}[T(A)]\right).\end{aligned} \tag{9}$$

Therefore Assumption 7 can be directly satisfied by assuming that $\psi_w$ is twice continuously differentiable *w.r.t.* $\mathbf{w}$.

**Assumption 8.** *For every $\theta \in \Theta$, $s \in \mathcal{S}$ and $\mathbf{w} \in W$, $f_\theta^\rho(\mathbf{w};s)$ (from Eq. equation 5) exists and is unique.*

The above assumption ensures that the true $(1-\rho)$-quantile is unique and the assumption is usually satisfied for most distributions and a well-behaved $Q_\theta$.

---

[4]For a distribution in NEF, there may exist multiple representations of the form (8). However, for the distribution, there definitely exists a representation where the components of the sufficient statistic are linearly independent and such a representation is referred to as *minimal*.

### B.3 MAIN THEOREM

To analyze the algorithm, we employ here the ODE-based analysis as proposed in (Borkar, 2008; Kushner & Clark, 2012). The actor recursions (Eqs. (6-7)) represent a classical two timescale stochastic approximation recursion, where there exists a bilateral coupling between the individual stochastic recursions (6) and (7). Since the step-size schedules $\{\alpha_{a,t}\}_{t\in\mathbb{N}}$ and $\{\alpha'_{a,t}\}_{t\in\mathbb{N}}$ satisfy $\frac{\alpha'_{a,t}}{\alpha_{a,t}} \to 0$, we have $\alpha'_{a,t} \to 0$ relatively faster than $\alpha_{a,t} \to 0$. This disparity induces a pseudo-heterogeneous rate of convergence (or timescales) between the individual stochastic recursions which further amounts to the asymptotic emergence of a stable coherent behaviour which is quasi-asynchronous. This pseudo-behaviour can be interpreted using multiple viewpoints. When viewed from the faster timescale recursion— controlled by $\alpha_{a,t}$—the slower timescale recursion—controlled by $\alpha'_{a,t}$—appears quasi-static, i.e., almost a constant. Likewise, when observed from the slower timescale, the faster timescale recursion seems equilibrated.

The existence of this stable long run behaviour under certain standard assumptions of stochastic approximation algorithms is rigorously established in (Borkar, 1997) and also in Chapter 6 of (Borkar, 2008). For our stochastic approximation setting (Eqs. (6-7)), we can directly apply this appealing characterization of the long run behaviour of the two timescale stochastic approximation algorithms—after ensuring the compliance of our setting to the pre-requisites demanded by the characterization—by considering the slow timescale stochastic recursion (7) to be quasi-stationary (i.e., $\mathbf{w}'_t \equiv \mathbf{w}'$, a.s., $\forall t \in \mathbb{N}$), while analyzing the limiting behaviour of the faster timescale recursion (6). Similarly, we let $\theta_t$ to be quasi-stationary too (i.e., $\theta_t \equiv \theta$, a.s., $\forall t \in \mathbb{N}$). The asymptotic behaviour of the slower timescale recursion is further analyzed by considering the faster timescale temporal variable $\mathbf{w}_t$ with the limit point so obtained during quasi-stationary analysis.

Define the filtration $\{\mathcal{F}_t\}_{t\in\mathbb{N}}$, a family of increasing natural $\sigma$-fields, where

$$\mathcal{F}_t \doteq \sigma\left(\{\mathbf{w}_i, \mathbf{w}'_i, (S_i, A_i, R_i, S'_i), \Xi_i; 0 \le i \le t\}\right).$$

**Theorem B.1.** *Let $\mathbf{w}'_t \equiv \mathbf{w}', \theta_t \equiv \theta, \forall t \in \mathbb{N}$ a.s. Let Assumptions 1-8 hold. Then the stochastic sequence $\{\mathbf{w}_t\}_{t\in\mathbb{N}}$ generated by the stochastic recursion (6) asymptotically tracks the ODE:*

$$\frac{d}{dt}\mathbf{w}(t) = \widehat{\Gamma}^W_{\mathbf{w}(t)}\left(\nabla_{\mathbf{w}(t)}\mathbb{E}_{S\sim\nu, A\sim\pi_{\mathbf{w}'}(\cdot|S)}\left[I_{\{Q_\theta(S,A)\ge f^\rho_\theta(\mathbf{w}';S)\}}\ln\pi_{\mathbf{w}(t)}(A|S)\right]\right), \quad t \ge 0. \quad (10)$$

*In other words, $\lim_{t\to\infty}\mathbf{w}_t \in \mathcal{K}$ a.s., where $\mathcal{K}$ is set of stable equilibria of the ODE (10) contained inside $W$.*

*Proof.* Firstly, we rewrite the stochastic recursion (6) under the hypothesis that $\theta_t$ and $\mathbf{w}'_t$ are quasi-stationary, i.e., $\theta_t \underset{a.s.}{\equiv} \theta$ and $\mathbf{w}'_t \underset{a.s.}{\equiv} \mathbf{w}'$ as follows:

$$\mathbf{w}_{t+1} \doteq \Gamma^W\left\{\mathbf{w}_t + \alpha_{a,t}\frac{1}{N_t}\sum_{A\in\Xi_t} I_{\{Q_\theta(S_t,A)\ge\widehat{f}^\rho_{t+1}\}}\nabla_{\mathbf{w}}\ln\pi_{\mathbf{w}}(A|S_t)\right\} \quad (11)$$

where $f^\rho_\theta(\mathbf{w}'; S) \doteq f^\rho(Q_\theta(S,\cdot), \pi_{\mathbf{w}'}(\cdot|S))$ and $\nabla_{\mathbf{w}_t} \doteq \nabla_{\mathbf{w}=\mathbf{w}_t}$, i.e., the gradient *w.r.t.* $\mathbf{w}$ at $\mathbf{w}_t$. Define

$$g^\theta(\mathbf{w}) \doteq \mathbb{E}_{S_t\sim\nu, A\sim\pi_{\mathbf{w}'}(\cdot|S_t)}\left[I_{\{Q_\theta(S_t,A)\ge f^\rho_\theta(\mathbf{w}';S_t)\}}\nabla_{\mathbf{w}}\ln\pi_{\mathbf{w}}(A|S_t)\right]. \quad (12)$$

$$\mathbb{M}_{t+1} \doteq \frac{1}{N_t}\sum_{A\in\Xi_t} I_{\{Q_\theta(S_t,A)\ge\widehat{f}^\rho_{t+1}\}}\nabla_{\mathbf{w}_t}\ln\pi_{\mathbf{w}}(A|S_t)-$$

$$\mathbb{E}\left[\frac{1}{N_t}\sum_{A\in\Xi_t} I_{\{Q_\theta(S_t,A)\ge\widehat{f}^\rho_{t+1}\}}\nabla_{\mathbf{w}_t}\ln\pi_{\mathbf{w}}(A|S_t)\Big|\mathcal{F}_t\right]. \quad (13)$$

$$\ell^\theta_t \doteq \mathbb{E}\left[\frac{1}{N_t}\sum_{A\in\Xi_t} I_{\{Q_\theta(S_t,A)\ge\widehat{f}^\rho_{t+1}\}}\nabla_{\mathbf{w}_t}\ln\pi_{\mathbf{w}}(A|S_t)\Big|\mathcal{F}_t\right]-$$

$$\mathbb{E}_{S_t\sim\nu, A\sim\pi_{\mathbf{w}'}(\cdot|S_t)}\left[I_{\{Q_\theta(S_t,A)\ge f^\rho_\theta(\mathbf{w}';S_t)\}}\nabla_{\mathbf{w}_t}\ln\pi_{\mathbf{w}}(A|S_t)\right] \quad (14)$$

Then we can rewrite

$$
\begin{aligned}
equation\ 11 = \Gamma^W \Bigg\{ \mathbf{w}_t + \alpha_{a,t} \Bigg( &\mathbb{E}_{S_t \sim \nu, A \sim \pi_{\mathbf{w}'}(\cdot|S_t)} \Big[ I_{\{Q_\theta(S_t,A) \geq f_\theta^\rho(\mathbf{w}';S_t)\}} \nabla_{\mathbf{w}_t} \ln \pi_{\mathbf{w}}(A|S_t) \Big] - \\
&\mathbb{E}_{S_t \sim \nu, A \sim \pi_{\mathbf{w}'}(\cdot|S_t)} \Big[ I_{\{Q_\theta(S_t,A) \geq f_\theta^\rho(\mathbf{w}';S_t)\}} \nabla_{\mathbf{w}_t} \ln \pi_{\mathbf{w}}(A|S_t) \Big] + \\
&\mathbb{E}\Big[ \frac{1}{N_t} \sum_{A \in \Xi_t} I_{\{Q_\theta(S_t,A) \geq \widehat{f}_{t+1}^\rho\}} \nabla_{\mathbf{w}_t} \ln \pi_{\mathbf{w}}(A|S_t) \Big| \mathscr{F}_t \Big] - \\
&\mathbb{E}\Big[ \frac{1}{N_t} \sum_{A \in \Xi_t} I_{\{Q_\theta(S_t,A) \geq \widehat{f}_{t+1}^\rho\}} \nabla_{\mathbf{w}_t} \ln \pi_{\mathbf{w}}(A|S_t) \Big| \mathscr{F}_t \Big] + \\
&\frac{1}{N_t} \sum_{A \in \Xi_t} I_{\{Q_\theta(S_t,A) \geq \widehat{f}_{t+1}^\rho\}} \nabla_{\mathbf{w}_t} \ln \pi_{\mathbf{w}}(A|S_t) \Bigg) \Bigg\}.
\end{aligned}
$$

$$
= \Gamma^W \Big\{ g^\theta(\mathbf{w}_t) + \mathbb{M}_{t+1} + \ell_t^\theta \Big\}, \tag{15}
$$

A few observations are in order:

B1. $\{\mathbb{M}_{t+1}\}_{t \in \mathbb{N}}$ is a martingale difference noise sequence *w.r.t.* the filtration $\{\mathscr{F}_t\}_{t \in \mathbb{N}}$, *i.e.*, $\mathbb{M}_{t+1}$ is $\mathscr{F}_{t+1}$-measurable and integrable, $\forall t \in \mathbb{N}$ and $\mathbb{E}[\mathbb{M}_{t+1}|\mathscr{F}_t] = 0$ *a.s.*, $\forall t \in \mathbb{N}$.

B2. $g^\theta$ is locally Lipschitz continuous. This follows from Assumption 7.

B3. $\ell_t^\theta \to 0$ *a.s.* as $t \to \infty$. (By Lemma 2 below).

B4. The iterates $\{\mathbf{w}_t\}_{t \in \mathbb{N}}$ is bounded almost surely, *i.e.*,

$$
\sup_{t \in \mathbb{N}} \|\mathbf{w}_t\| < \infty \quad a.s.
$$

This is ensured by the explicit application of the projection operator $\Gamma^W\{\cdot\}$ over the iterates $\{\mathbf{w}_t\}_{t \in \mathbb{N}}$ at every iteration onto the bounded set $W$.

B5. $\exists L \in (0, \infty)$ *s.t.* $\mathbb{E}\big[ \|\mathbb{M}_{t+1}\|^2 | \mathscr{F}_t \big] \leq L \big( 1 + \|\mathbf{w}_t\|^2 \big)$ *a.s.*

This follows from Assumption 6 (ii).

Now, we rewrite the stochastic recursion (15) as follows:

$$
\begin{aligned}
\mathbf{w}_{t+1} &\doteq \mathbf{w}_t + \alpha_{a,t} \frac{\Gamma^W \big\{ \mathbf{w}_t + \xi_t \big( g^\theta(\mathbf{w}_t) + \mathbb{M}_{t+1} + \ell_t^\theta \big) \big\} - \mathbf{w}_t}{\alpha_{a,t}} \\
&= \mathbf{w}_t + \alpha_{a,t} \Big( \widehat{\Gamma}^W_{\mathbf{w}_t}(g^\theta(\mathbf{w}_t)) + \widehat{\Gamma}^W_{\mathbf{w}_t}(\mathbb{M}_{t+1}) + \widehat{\Gamma}^W_{\mathbf{w}_t}(\ell_t^\theta) + o(\alpha_{a,t}) \Big), \tag{16}
\end{aligned}
$$

where $\widehat{\Gamma}^W$ is the Frechet derivative (Definition 3).

The above stochastic recursion is also a stochastic approximation recursion with the vector field $\widehat{\Gamma}^W_{\mathbf{w}_t}(g^\theta(\mathbf{w}_t))$, the noise term $\widehat{\Gamma}^W_{\mathbf{w}_t}(\mathbb{M}_{t+1})$, the bias term $\widehat{\Gamma}^W_{\mathbf{w}_t}(\ell_t^\theta)$ with an additional error term $o(\alpha_{a,t})$ which is asymptotically inconsequential.

Also, note that $\Gamma^W$ is single-valued map since the set $W$ is assumed convex and also the limit exists since the boundary $\partial W$ is considered smooth. Further, for $\mathbf{w} \in \mathring{W}$, we have

$$
\widehat{\Gamma}^W_{\mathbf{w}}(\mathbf{u}) \doteq \lim_{\epsilon \to 0} \frac{\Gamma^W\{\mathbf{w} + \epsilon\mathbf{u}\} - \mathbf{w}}{\epsilon} = \lim_{\epsilon \to 0} \frac{\mathbf{w} + \epsilon\mathbf{u} - \mathbf{w}}{\epsilon} = \mathbf{u} \text{ (for sufficiently small } \epsilon), \tag{17}
$$

*i.e.*, $\widehat{\Gamma}_{\mathbf{w}}^W(\cdot)$ is an identity map for $\mathbf{w} \in \mathring{W}$.

Now by appealing to Theorem 2, Chapter 2 of (Borkar, 2008) along with the observations B1-B5, we conclude that the stochastic recursion (6) asymptotically tracks the following ODE almost surely:

$$
\begin{aligned}
\frac{d}{dt}\mathbf{w}(t) &= \widehat{\Gamma}_{\mathbf{w}(t)}^W(g^\theta(\mathbf{w}(t))), \quad t \geq 0 \\
&= \widehat{\Gamma}_{\mathbf{w}(t)}^W \left( \mathbb{E}_{S\sim\nu, A\sim\pi_{\mathbf{w}'}(\cdot|S)} \left[ I_{\{Q_\theta(S,A)\geq f_\theta^\rho(\mathbf{w}';S)\}} \nabla_{\mathbf{w}(t)} \ln \pi_{\mathbf{w}}(A|S) \right] \right) \\
&= \widehat{\Gamma}_{\mathbf{w}(t)}^W \left( \nabla_{\mathbf{w}(t)} \mathbb{E}_{S\sim\nu, A\sim\pi_{\mathbf{w}'}(\cdot|S)} \left[ I_{\{Q_\theta(S,A)\geq f_\theta^\rho(\mathbf{w}';S)\}} \ln \pi_{\mathbf{w}}(A|S) \right] \right).
\end{aligned} \tag{18}
$$

The interchange of expectation and the gradient in the last equality follows from dominated convergence theorem and Assumption 7 (Rubinstein & Shapiro, 1993). The above ODE is a gradient flow with dynamics restricted inside $W$. This further implies that the stochastic recursion (6) converges to a (possibly sample path dependent) asymptotically stable equilibrium point of the above ODE inside $W$. □

### B.4 PROOF OF LEMMA 2 TO SATISFY CONDITION 3

In this section, we show that $\ell_t^\theta \to 0$ *a.s.* as $t \to \infty$, in Lemma 2. To do so, we first need to prove several supporting lemmas. Lemma 1 shows that, for a given Actor and Expert, the sample quantile converges to the true quantile. Using this lemma, we can then prove Lemma 2. In the following subsection, we provide three supporting lemmas about convexity and Lipschitz properties of the sample quantiles, required for the proof Lemma 1.

For this section, we require the following characterization of $f^\rho(Q_\theta(s, \cdot), \mathbf{w}')$. Please refer Lemma 1 of (Homem-de Mello, 2007) for more details.

$$
f^\rho(Q_\theta(s, \cdot), \mathbf{w}') = \underset{\ell \in [Q_l^\theta, Q_u^\theta]}{\arg\min} \, \mathbb{E}_{A\sim\pi_{\mathbf{w}'}(\cdot|s)} [\Psi(Q_\theta(s, A), \ell)], \tag{19}
$$

where $\Psi(y, \ell) \doteq (y - \ell)(1 - \rho)I_{\{y\geq\ell\}} + (\ell - y)\rho I_{\{\ell\geq y\}}$.

Similarly, the sample estimate of the true $(1 - \rho)$-quantile, *i.e.*, $\widehat{f}^\rho \doteq Q_{\theta,s}^{(\lceil(1-\rho)N\rceil)}$, (where $Q_{\theta,s}^{(i)}$ is the $i$-th order statistic of the random sample $\{Q_\theta(s, A)\}_{A\in\Xi}$ with $\Xi \doteq \{A_i\}_{i=1}^N \overset{iid}{\sim} \pi_{\mathbf{w}'}(\cdot|s)$) can be characterized as the unique solution of the stochastic counterpart of the above optimization problem, *i.e.*,

$$
\widehat{f}^\rho = \underset{\ell \in [Q_l^\theta, Q_u^\theta]}{\arg\min} \, \frac{1}{N} \sum_{\substack{A\in\Xi \\ |\Xi|=N}} \Psi(Q_\theta(s, A), \ell). \tag{20}
$$

**Lemma 1.** *Assume $\theta_t \equiv \theta$, $\mathbf{w}_t' \equiv \mathbf{w}'$, $\forall t \in \mathbb{N}$. Also, let Assumptions 3-5 hold. Then, for a given state $s \in \mathcal{S}$,*

$$
\lim_{t\to\infty} \widehat{f}_t^\rho = f^\rho(Q_\theta(s, \cdot), \mathbf{w}') \ \ a.s.,
$$

*where $\widehat{f}_t^\rho \doteq Q_{\theta,s}^{(\lceil(1-\rho)N_t\rceil)}$, (where $Q_{\theta,s}^{(i)}$ is the $i$-th order statistic of the random sample $\{Q_\theta(s, A)\}_{A\in\Xi_t}$ with $\Xi_t \doteq \{A_i\}_{i=1}^{N_t} \overset{iid}{\sim} \pi_{\mathbf{w}'}(\cdot|s))$.*

*Proof.* The proof is similar to arguments in Lemma 7 of (Hu et al., 2007). Since state $s$ and expert parameter $\theta$ are considered fixed, we assume the following notation in the proof. Let

$$
\widehat{f}_{t|s,\theta}^\rho \doteq \widehat{f}_t^\rho \text{ and } f_{|s,\theta}^\rho \doteq f^\rho(Q_\theta(s, \cdot), \mathbf{w}'), \tag{21}
$$

where $\widehat{f}_t^\rho$ and $f^\rho(Q_\theta(s, \cdot), \mathbf{w}')$ are defined in Equations equation 19 and equation 20.

Consider the open cover $\{B_r(\ell), \ell \in [Q_l^\theta, Q_u^\theta]\}$ of $[Q_l^\theta, Q_u^\theta]$. Since $[Q_l^\theta, Q_u^\theta]$ is compact, there exists a finite sub-cover, *i.e.*, $\exists\{\ell_1, \ell_2, \ldots, \ell_M\}$ *s.t.* $\cup_{i=1}^M B_r(\ell_i) = [Q_l^\theta, Q_u^\theta]$. Let

$\vartheta(\ell) \doteq \mathbb{E}_{A \sim \pi_{\mathbf{w}'}(\cdot|S)} [\Psi(Q_\theta(s,A),\ell)]$ and $\widehat{\vartheta}_t(\ell) \doteq \frac{1}{N_t} \sum\limits_{\substack{A \in \Xi_t, |\Xi_t|=N_t, \\ \Xi_t \overset{iid}{\sim} \pi_{\mathbf{w}'}(\cdot|s)}} \Psi(Q_\theta(s,A),\ell).$

Now, by triangle inequality, we have for $\ell \in [Q_l^\theta, Q_u^\theta]$,

$$
\begin{aligned}
|\vartheta(\ell) - \widehat{\vartheta}_t(\ell)| &\le |\vartheta(\ell) - \vartheta(\ell_j)| + |\vartheta(\ell_j) - \widehat{\vartheta}_t(\ell_j)| + |\widehat{\vartheta}_t(\ell_j) - \widehat{\vartheta}_t(\ell)| \\
&\le L_\rho |\ell - \ell_j| + |\vartheta(\ell_j) - \widehat{\vartheta}_t(\ell_j)| + \widehat{L}_\rho |\ell_j - \ell| \\
&\le \left( L_\rho + \widehat{L}_\rho \right) r + |\vartheta(\ell_j) - \widehat{\vartheta}_t(\ell_j)|,
\end{aligned}
\tag{22}
$$

where $L_\rho$ and $\widehat{L}_\rho$ are the Lipschitz constants of $\vartheta(\cdot)$ and $\widehat{\vartheta}_t(\cdot)$ respectively.

For $\delta > 0$, take $r = \delta(L_\rho + \widehat{L}_\rho)/2$. Also, by Kolmogorov's strong law of large numbers (Theorem 2.3.10 of (Sen & Singer, 2017)), we have $\widehat{\vartheta}_t(\ell) \to \vartheta(\ell)$ a.s. This implies that there exists $T \in \mathbb{N}$ s.t. $|\vartheta(\ell_j) - \widehat{\vartheta}_t(\ell_j)| < \delta/2, \forall t \ge T, \forall j \in [M]$. Then from Eq. (22), we have

$$
|\vartheta(\ell) - \widehat{\vartheta}_t(\ell)| \le \delta/2 + \delta/2 = \delta, \quad \forall \ell \in [Q_l^\theta, Q_u^\theta].
$$

This implies $\widehat{\vartheta}_t$ converges uniformly to $\vartheta$. By Lemmas 3 and 4, $\widehat{\vartheta}_t$ and $\vartheta$ are strictly convex and Lipschitz continuous, and so because $\widehat{\vartheta}_t$ converges uniformly to $\vartheta$, this means that the sequence of minimizers of $\widehat{\vartheta}_t$ converge to the minimizer of $\vartheta$ (see Lemma 5, Appendix B.6 for an explicit justification). These minimizers correspond to $\widehat{f}_t^\rho$ and $f^\rho(Q_\theta(s,\cdot),\mathbf{w}')$ respectively, and so $\lim_{N_t \to \infty} \widehat{f}_t^\rho = f^\rho(Q_\theta(s,\cdot),\mathbf{w}')$ a.s.

Now, for $\delta > 0$ and $r \doteq \delta(L_\rho + \widehat{L}_\rho)/2$, we obtain the following from Eq. (22):

$$
\begin{aligned}
&|\vartheta(\ell) - \widehat{\vartheta}_t(\ell)| \le \delta/2 + |\vartheta(\ell_j) - \widehat{\vartheta}_t(\ell_j)| \\
&\Leftrightarrow \{|\vartheta(\ell_j) - \widehat{\vartheta}_t(\ell_j)| \le \delta/2, \forall j \in [M]\} \Rightarrow \{|\vartheta(\ell) - \widehat{\vartheta}_t(\ell)| \le \delta, \forall \ell \in [Q_l^\theta, Q_u^\theta]\}
\end{aligned}
$$

$$
\begin{aligned}
\Rightarrow \mathbb{P}_{\pi_{\mathbf{w}'}} \left( |\vartheta(\ell) - \widehat{\vartheta}_t(\ell)| \le \delta, \forall \ell \in [Q_l^\theta, Q_u^\theta] \right) &\ge \mathbb{P}_{\pi_{\mathbf{w}'}} \left( |\vartheta(\ell_j) - \widehat{\vartheta}_t(\ell_j)| \le \delta/2, \forall j \in [M] \right) \\
&= 1 - \mathbb{P}_{\pi_{\mathbf{w}'}} \left( |\vartheta(\ell_j) - \widehat{\vartheta}_t(\ell_j)| > \delta/2, \exists j \in [M] \right) \\
&\ge 1 - \sum_{j=1}^{M} \mathbb{P}_{\pi_{\mathbf{w}'}} \left( |\vartheta(\ell_j) - \widehat{\vartheta}_t(\ell_j)| > \delta/2 \right) \\
&\ge 1 - M \max_{j \in [M]} \mathbb{P}_{\pi_{\mathbf{w}'}} \left( |\vartheta(\ell_j) - \widehat{\vartheta}_t(\ell_j)| > \delta/2 \right) \\
&\ge 1 - 2M \exp \left( \frac{-2N_t \delta^2}{4(Q_u^\theta - Q_l^\theta)^2} \right),
\end{aligned}
\tag{23}
$$

where $\mathbb{P}_{\pi_{\mathbf{w}'}} \doteq \mathbb{P}_{A \sim \pi_{\mathbf{w}'}}(\cdot|s)$. And the last inequality follows from Hoeffding's inequality (Hoeffding, 1963) along with the fact that $\mathbb{E}_{\pi_{\mathbf{w}'}} \left[ \widehat{\vartheta}_t(\ell_j) \right] = \vartheta(\ell_j)$ and $\sup\limits_{\ell \in [Q_l^\theta, Q_u^\theta]} |\vartheta(\ell)| \le Q_u^\theta - Q_l^\theta$.

Now, the sub-differential of $\vartheta(\ell)$ is given by

$$
\partial_\ell \vartheta(\ell) = \left[ \rho - \mathbb{P}_{A \sim \pi_{\mathbf{w}'}(\cdot|s)} \left( Q_\theta(s,A) \ge \ell \right), \rho - 1 + \mathbb{P}_{A \sim \pi_{\mathbf{w}'}(\cdot|s)} \left( Q_\theta(s,A) \le \ell \right) \right].
\tag{24}
$$

By the definition of sub-gradient we obtain

$$
\begin{aligned}
&c|\widehat{f}_{t|s,\theta}^\rho - f_{|s,\theta}^\rho| \le |\vartheta(\widehat{f}_{t|s,\theta}^\rho) - \vartheta(f_{|s,\theta}^\rho)|, \quad c \in \partial_\ell \vartheta(\ell) \\
&\Rightarrow C|\widehat{f}_{t|s,\theta}^\rho - f_{|s,\theta}^\rho| \le |\vartheta(\widehat{f}_{t|s,\theta}^\rho) - \vartheta(f_{|s,\theta}^\rho)|,
\end{aligned}
\tag{25}
$$

where $C \doteq \max \left\{ \rho - \mathbb{P}_{A \sim \pi_{\mathbf{w}'}(\cdot|s)} \left( Q_\theta(s, A) \geq f^\rho_{|s,\theta} \right), \rho - 1 + \mathbb{P}_{A \sim \pi_{\mathbf{w}'}(\cdot|s)} \left( Q_\theta(s, A) \leq f^\rho_{|s,\theta} \right) \right\}$.
Further,

$$
\begin{aligned}
C|\widehat{f}^\rho_{t|s,\theta} - f^\rho_{|s,\theta}| &\leq |\vartheta(\widehat{f}^\rho_{t|s,\theta}) - \vartheta(f^\rho_{|s,\theta})| \\
&\leq |\vartheta(\widehat{f}^\rho_{t|s,\theta}) - \widehat{\vartheta}_t(\widehat{f}^\rho_{t|s,\theta})| + |\widehat{\vartheta}_t(\widehat{f}^\rho_{t|s,\theta}) - \vartheta(f^\rho_{|s,\theta})| \\
&\leq |\vartheta(\widehat{f}^\rho_{t|s,\theta}) - \widehat{\vartheta}_t(\widehat{f}^\rho_{t|s,\theta})| + \sup_{\ell \in [Q^\theta_l, Q^\theta_u]} |\widehat{\vartheta}_t(\ell) - \vartheta(\ell)| \\
&\leq 2 \sup_{\ell \in [Q^\theta_l, Q^\theta_u]} |\widehat{\vartheta}_t(\ell) - \vartheta(\ell)|.
\end{aligned}
\tag{26}
$$

From Eqs. (23) and (26), we obtain for $\epsilon > 0$

$$
\begin{aligned}
\mathbb{P}_{\mathbf{w}'} \left( N^\alpha_t |\widehat{f}^\rho_{t|s,\theta} - f^\rho_{|s,\theta}| \geq \epsilon \right) &\leq \mathbb{P}_{\mathbf{w}'} \left( N^\alpha_t \sup_{\ell \in [Q^\theta_l, Q^\theta_u]} |\widehat{\vartheta}_t(\ell) - \vartheta(\ell)| \geq \frac{\epsilon}{2} \right) \\
&\leq 2M \exp \left( \frac{-2N_t \epsilon^2}{16 N^{2\alpha}_t (Q^\theta_u - Q^\theta_l)^2} \right) = 2M \exp \left( \frac{-2N^{1-2\alpha}_t \epsilon^2}{16 (Q^\theta_u - Q^\theta_l)^2} \right).
\end{aligned}
$$

For $\alpha \in (0, 1/2)$ and $\inf_{t \in \mathbb{N}} \frac{N_{t+1}}{N_t} \geq \tau > 1$ (by Assumption 3), then

$$
\sum_{t=1}^\infty 2M \exp \left( \frac{-2N^{1-2\alpha}_t \epsilon^2}{16 (Q^\theta_u - Q^\theta_l)^2} \right) \leq \sum_{t=1}^\infty 2M \exp \left( \frac{-2\tau^{(1-2\alpha)t} N^{1-2\alpha}_0 \epsilon^2}{16 (Q^\theta_u - Q^\theta_l)^2} \right) < \infty.
$$

Therefore, by Borel-Cantelli's Lemma (Durrett, 1991), we have

$$
\mathbb{P}_{\mathbf{w}'} \left( N^\alpha_t |\widehat{f}^\rho_{t|s,\theta} - f^\rho_{|s,\theta}| \geq \epsilon \ \ i.o \right) = 0.
$$

Thus we have $N^\alpha_t \left( \widehat{f}^\rho_{t|s,\theta} - f^\rho_{|s,\theta} \right) \to 0$ $a.s.$ as $N_t \to \infty$. $\qquad \square$

**Lemma 2.** *Almost surely,*

$$
\ell^\theta_t \to 0 \quad as \ N_t \to \infty.
$$

**Proof of Lemma 2:** Consider

$$
\mathbb{E} \left[ \frac{1}{N_t} \sum_{A \in \Xi_t} I_{\{Q_\theta(S_t, A) \geq \widehat{f}^\rho_{t+1}\}} \nabla_{\mathbf{w}_t} \ln \pi_{\mathbf{w}}(A|S_t) \middle| \mathscr{F}_t \right] =
$$

$$
\mathbb{E} \left[ \mathbb{E}_{\Xi_t} \left[ \frac{1}{N_t} \sum_{A \in \Xi_t} I_{\{Q_\theta(S_t, A) \geq \widehat{f}^\rho_{t+1}\}} \nabla_{\mathbf{w}_t} \ln \pi_{\mathbf{w}}(A|S_t) \right] \middle| S_t = s, \mathbf{w}'_t \right]
$$

Notice that, because of the conditions on $\pi_{\mathbf{w}'}(\cdot|s)$, we know that the sample average converges with an exponential rate in the number of samples, for arbitrary $\mathbf{w}' \in W$. Namely, for $\epsilon > 0$ and $N \in \mathbb{N}$, we have

$$
\mathbb{P}_{\Xi \overset{\text{iid}}{\sim} \pi_{\mathbf{w}'}(\cdot|s)} \left( \left\| \frac{1}{N} \sum_{A \in \Xi} I_{\{Q_\theta(s,A) \geq f^\rho(Q_\theta(s,\cdot), \pi_{\mathbf{w}'}(\cdot|s))\}} \nabla_{\mathbf{w}} \ln \pi_{\mathbf{w}}(A|s) - \right.\right.
$$

$$
\mathbb{E}_{A \sim \pi_{\mathbf{w}'}(\cdot|s)} \left[ I_{\{Q_\theta(s,A) \geq \widehat{f}^\rho(Q_\theta(s,\cdot), \pi_{\mathbf{w}'}(\cdot|s))\}} \nabla_{\mathbf{w}} \ln \pi_{\mathbf{w}}(A|s) \right] \Bigg\| \geq \epsilon \right) \leq C_1 \exp \left( -c_2 N^{c_3} \epsilon^{c_4} \right),
$$

$$
\forall \theta \in \Theta, \mathbf{w}, \mathbf{w}' \in W, s \in \mathscr{S},
$$

where $C_1, c_2, c_3, c_4 > 0$.

Therefore, for $\alpha' > 0$, we have

$$
\mathbb{P} \left( N^{\alpha'}_t \left\| \frac{1}{N_t} \sum_{A \in \Xi_t} I_{\{Q_\theta(s,A) \geq \widehat{f}^\rho_{\theta,s}\}} \nabla_{\mathbf{w}_t} \ln \pi_{\mathbf{w}}(A|s) - \mathbb{E} \left[ I_{\{Q_\theta(s,A) \geq \widehat{f}^\rho_{\theta,s}\}} \nabla_{\mathbf{w}_t} \ln \pi_{\mathbf{w}}(A|s) \right] \right\| \geq \epsilon \right)
$$

$$
\leq C_1 \exp \left( -\frac{c_2 N^{c_3}_t \epsilon^{c_4}}{N^{c_4 \alpha'}_t} \right) = C_1 \exp \left( -c_2 N^{c_3 - c_4 \alpha'}_t \epsilon^{c_4} \right)
$$

$$
\leq C_1 \exp \left( -c_2 \tau^{(c_3 - c_4 \alpha')t} N^{c_3 - c_4 \alpha'}_0 \epsilon^{c_4} \right),
$$

where $f^\rho_{\theta,s} \doteq f^\rho(Q_\theta(s,\cdot), \pi_{\mathbf{w}'}(\cdot|s))$ and $\inf_{t\in\mathbb{N}} \frac{N_{t+1}}{N_t} \geq \tau > 1$ (by Assumption 3).

For $c_3 - c_4\alpha' > 0 \Rightarrow \alpha' < c_3/c_4$, we have

$$\sum_{t=1}^{\infty} C_1 \exp\left(-c_2\tau^{(c_3-c_4\alpha')t}N_0^{c_3-c_4\alpha'}\epsilon^{c_4}\right) < \infty.$$

Therefore, by Borel-Cantelli's Lemma (Durrett, 1991), we have

$$\mathbb{P}\left(N_t^{\alpha'}\Big\|\frac{1}{N_t}\sum_{A\in\Xi_t} I_{\{Q_\theta(s,A)\geq \widehat{f}^\rho_{\theta,s}\}}\nabla_{\mathbf{w}_t}\ln\pi_{\mathbf{w}}(A|s) - \mathbb{E}\left[I_{\{Q_\theta(s,A)\geq f^\rho_{\theta,s}\}}\nabla_{\mathbf{w}_t}\ln\pi_{\mathbf{w}}(A|s)\right]\Big\| \geq \epsilon \; i.o.\right)$$
$$= 0.$$

This implies that

$$N_t^{\alpha'}\Big\|\frac{1}{N_t}\sum_{A\in\Xi_t} I_{\{Q_\theta(s,A)\geq \widehat{f}^\rho_{\theta,s}\}}\nabla_{\mathbf{w}_t}\ln\pi_{\mathbf{w}}(A|s) - \mathbb{E}\left[I_{\{Q_\theta(s,A)\geq \widehat{f}^\rho_{\theta,s}\}}\nabla_{\mathbf{w}_t}\ln\pi_{\mathbf{w}}(A|s)\right]\Big\| \to 0 \quad a.s. \tag{27}$$

The above result implies that the sample average converges at a rate $O(N_t^{\alpha'})$, where $0 < \alpha' < c_3/c_4$ independent of $\mathbf{w}, \mathbf{w}' \in W$. By Lemma 1, we have the sample quantiles $\widehat{f}^\rho_t$ also converging to the true quantile at a rate $O(N_t^{\alpha})$ independent of $\mathbf{w}, \mathbf{w}' \in W$. Now the claim follows directly from Assumption 6 (ii) and bounded convergence theorem. ∎

## B.5 SUPPORTING LEMMAS FOR LEMMA 1

**Lemma 3.** *Let Assumption 5 hold. For $\theta \in \Theta$, $\mathbf{w}' \in W$, $s \in \mathcal{S}$ and $\ell \in [Q_l^\theta, Q_u^\theta]$, we have*

1. $\mathbb{E}_{A\sim\pi_{\mathbf{w}'}(\cdot|s)}\left[\Psi(Q_\theta(s,A),\ell)\right]$ *is Lipschitz continuous.*

2. $\frac{1}{N}\sum_{\substack{A\in\Xi \\ |\Xi|=N}}\Psi(Q_\theta(s,A),\ell)$ *(with $\Xi \stackrel{iid}{\sim} \pi_{\mathbf{w}'}(\cdot|s)$) is Lipschitz continuous with Lipschitz constant independent of the sample length $N$.*

*Proof.* Let $\ell_1, \ell_2 \in [Q_l^\theta, Q_u^\theta]$, $\ell_2 \geq \ell_1$. By Assumption 5 we have $\mathbb{P}_{A\sim\pi_{\mathbf{w}'}(\cdot|s)}(Q_\theta(s,A) \geq \ell_1) > 0$ and $\mathbb{P}_{A\sim\pi_{\mathbf{w}'}(\cdot|s)}(Q_\theta(s,A) \geq \ell_2) > 0$. Now,

$$\left|\mathbb{E}_{A\sim\pi_{\mathbf{w}'}(\cdot|s)}\left[\Psi(Q_\theta(s,A),\ell_1)\right] - \mathbb{E}_{A\sim\pi_{\mathbf{w}'}(\cdot|s)}\left[\Psi(Q_\theta(s,A),\ell_2)\right]\right|$$

$$= \Big|\mathbb{E}_{A\sim\pi_{\mathbf{w}'}(\cdot|s)}\left[(Q_\theta(s,A)-\ell_1)(1-\rho)I_{\{Q_\theta(s,A)\geq\ell_1\}} + (\ell_1-Q_\theta(s,A))\rho I_{\{\ell_1\geq Q_\theta(s,A)\}}\right]$$

$$- \mathbb{E}_{A\sim\pi_{\mathbf{w}'}(\cdot|s)}\left[(Q_\theta(s,A)-\ell_2)(1-\rho)I_{\{Q_\theta(s,A)\geq\ell_2\}} + (\ell_2-Q_\theta(s,A))\rho I_{\{\ell_2\geq Q_\theta(s,A)\}}\right]\Big|$$

$$= \Big|\mathbb{E}_{A\sim\pi_{\mathbf{w}'}(\cdot|s)}\Big[(Q_\theta(s,A)-\ell_1)(1-\rho)I_{\{Q_\theta(s,A)\geq\ell_1\}} + (\ell_1-Q_\theta(s,A))\rho I_{\{\ell_1\geq Q_\theta(s,A)\}}$$

$$- (Q_\theta(s,A)-\ell_2)(1-\rho)I_{\{Q_\theta(s,A)\geq\ell_2\}} + (\ell_2-Q_\theta(s,A))\rho I_{\{\ell_2\geq Q_\theta(s,A)\}}\Big]\Big|$$

$$= \Big|\mathbb{E}_{A\sim\pi_{\mathbf{w}'}(\cdot|s)}\Big[(1-\rho)(\ell_2-\ell_1)I_{\{Q_\theta(s,A)\geq\ell_2\}} + \rho(\ell_1-\ell_2)I_{\{Q_\theta(s,A)\leq\ell_1\}}+$$

$$+ (-(1-\rho)\ell_1 - \rho\ell_2 + \rho Q_\theta(s,A) + (1-\rho)Q_\theta(s,A))I_{\{\ell_1\leq Q_\theta(s,A)\leq\ell_2\}}\Big]\Big|$$

$$\leq (1-\rho)|\ell_2-\ell_1| + (2\rho+1)|\ell_2-\ell_1|$$

$$= (\rho+2)|\ell_2-\ell_1|.$$

Similarly, we can prove the later claim also. This completes the proof of Lemma 3. □

**Lemma 4.** *Let Assumption 5 hold. Then, for $\theta \in \Theta$, $\mathbf{w}' \in W$, $s \in \mathcal{S}$ and $\ell \in [Q_l^\theta, Q_u^\theta]$, we have $\mathbb{E}_{A\sim\pi_{\mathbf{w}'}(\cdot|s)}\left[\Psi(Q_\theta(s,A),\ell)\right]$ and $\frac{1}{N}\sum_{\substack{A\in\Xi \\ |\Xi|=N}}\Psi(Q_\theta(s,A),\ell)$ (with $\Xi \stackrel{iid}{\sim} \pi_{\mathbf{w}'}(\cdot|s)$) are strictly convex.*

*Proof.* For $\lambda \in [0,1]$ and $\ell_1, \ell_2 \in [Q_l, Q_u]$ with $\ell_1 \leq \ell_2$, we have

$$\mathbb{E}_{A \in \pi_{\mathbf{w}'}(\cdot|S)}\big[\Psi(Q_\theta(S,A), \lambda\ell_1 + (1-\lambda)\ell_2)\big] \tag{28}$$
$$= \mathbb{E}_{A \in \pi_{\mathbf{w}'}(\cdot|S)}\big[(1-\rho)\big(Q_\theta(S,A) - \lambda\ell_1 - (1-\lambda)\ell_2\big)I_{\{Q_\theta(S,A) \geq \lambda\ell_1 + (1-\lambda)\ell_2\}}$$
$$+ \rho\big(\lambda\ell_1 + (1-\lambda)\ell_2 - Q_\theta(S,A)\big)I_{\{Q_\theta(S,A) \leq \lambda\ell_1 + (1-\lambda)\ell_2\}}\big].$$

Notice that

$$\big(Q_\theta(S,A) - \lambda\ell_1 - (1-\lambda)\ell_2\big)I_{\{Q_\theta(S,A) \geq \lambda\ell_1 + (1-\lambda)\ell_2\}}$$
$$= \big(\lambda Q_\theta(S,A) - \lambda\ell_1 + (1-\lambda)Q_\theta(S,A) - (1-\lambda)\ell_2\big)I_{\{Q_\theta(S,A) \geq \lambda\ell_1 + (1-\lambda)\ell_2\}}$$

We consider how one of these components simplifies.

$$\mathbb{E}_{A \in \pi_{\mathbf{w}'}(\cdot|S)}\big[\big(\lambda Q_\theta(S,A) - \lambda\ell_1\big)I_{\{Q_\theta(S,A) \geq \lambda\ell_1 + (1-\lambda)\ell_2\}}\big]$$
$$= \lambda\mathbb{E}_{A \in \pi_{\mathbf{w}'}(\cdot|S)}\big[\big(Q_\theta(S,A) - \ell_1\big)I_{\{Q_\theta(S,A) \geq \lambda\ell_1\}} - \big(Q_\theta(S,A) - \ell_1\big)I_{\lambda\ell_1 \leq \{Q_\theta(S,A) \leq \lambda\ell_1 + (1-\lambda)\ell_2\}}\big]$$
$$\leq \lambda\mathbb{E}_{A \in \pi_{\mathbf{w}'}(\cdot|S)}\big[\big(Q_\theta(S,A) - \ell_1\big)I_{\{Q_\theta(S,A) \geq \lambda\ell_1\}}\big] \quad \triangleright \; -\big(Q_\theta(S,A) - \ell_1\big) \leq 0$$
$$\text{for } \lambda\ell_1 \leq \{Q_\theta(S,A) \leq \lambda\ell_1 + (1-\lambda)\ell_2\}$$
$$\leq \lambda\mathbb{E}_{A \in \pi_{\mathbf{w}'}(\cdot|S)}\big[\big(Q_\theta(S,A) - \ell_1\big)I_{\{Q_\theta(S,A) \geq \ell_1\}}\big] \quad \triangleright \; \big(Q_\theta(S,A) - \ell_1\big) \leq 0 \text{ for } I_{\lambda\ell_1 \leq \{Q_\theta(S,A) \leq \ell_1\}}$$

Similarly, we get

$$\mathbb{E}_{A \in \pi_{\mathbf{w}'}(\cdot|S)}\big[\big(Q_\theta(S,A) - \ell_2\big)I_{\{Q_\theta(S,A) \geq \lambda\ell_1 + (1-\lambda)\ell_2\}}\big] \leq \mathbb{E}_{A \in \pi_{\mathbf{w}'}(\cdot|S)}\big[\big(Q_\theta(S,A) - \ell_2\big)I_{\{Q_\theta(S,A) \geq \ell_2\}}\big]$$
$$\mathbb{E}_{A \in \pi_{\mathbf{w}'}(\cdot|S)}\big[\big(\ell_1 - Q_\theta(S,A)\big)I_{\{Q_\theta(S,A) \leq \lambda\ell_1 + (1-\lambda)\ell_2\}}\big] \leq \mathbb{E}_{A \in \pi_{\mathbf{w}'}(\cdot|S)}\big[\big(\ell_1 - Q_\theta(S,A)\big)I_{\{Q_\theta(S,A) \leq \ell_1\}}\big]$$
$$\mathbb{E}_{A \in \pi_{\mathbf{w}'}(\cdot|S)}\big[\big(\ell_2 - Q_\theta(S,A)\big)I_{\{Q_\theta(S,A) \leq \lambda\ell_1 + (1-\lambda)\ell_2\}}\big] \leq \mathbb{E}_{A \in \pi_{\mathbf{w}'}(\cdot|S)}\big[\big(\ell_2 - Q_\theta(S,A)\big)I_{\{Q_\theta(S,A) \leq \ell_2\}}\big]$$

Therefore, for Equation equation 28, we get

$$equation\ 28 \leq \lambda(1-\rho)\mathbb{E}_{A \in \pi_{\mathbf{w}'}(\cdot|S)}\big[\big(Q_\theta(S,A) - \ell_1\big)I_{\{Q_\theta(S,A) \geq \ell_1\}}\big]$$
$$+ (1-\lambda)(1-\rho)\mathbb{E}_{A \in \pi_{\mathbf{w}'}(\cdot|S)}\big[\big(Q_\theta(S,A) - \ell_2\big)I_{\{Q_\theta(S,A) \geq \ell_2\}}\big]$$
$$+ \lambda\rho\mathbb{E}_{A \in \pi_{\mathbf{w}'}(\cdot|S)}\big[\big(\ell_1 - Q_\theta(S,A)\big)I_{\{Q_\theta(S,A) \leq \ell_1\}}\big]$$
$$+ (1-\lambda)\rho\mathbb{E}_{A \in \pi_{\mathbf{w}'}(\cdot|S)}\big[\big(\ell_2 - Q_\theta(S,A)\big)I_{\{Q_\theta(S,A) \leq \ell_2\}}\big]$$
$$= \lambda\mathbb{E}_{A \in \pi_{\mathbf{w}'}(\cdot|S)}\big[\Psi(Q_\theta(S,A), \ell_1)\big] + (1-\lambda)\mathbb{E}_{A \in \pi_{\mathbf{w}'}(\cdot|S)}\big[\Psi(Q_\theta(S,A), \ell_2)\big].$$

We can prove the second claim similarly. This completes the proof of Lemma 4. $\qquad\square$

### B.6   LEMMA 5

**Lemma 5.** *Let $\{f_n \in C(\mathbb{R}, \mathbb{R})\}_{n \in \mathbb{N}}$ be a sequence of strictly convex, continuous functions converging uniformly to a strict convex function $f$. Let $x_n^* = \arg\min_x f_n(x)$ and $x^* = \arg\min_{x \in \mathbb{R}} f(x)$. Then $\lim_{n \to \infty} x_n^* = x^*$.*

*Proof.* Let $c = \liminf_n x_n^*$. We employ proof by contradiction here. For that, we assume $x^* > c$. Now, note that $f(x^*) < f(c)$ and $f(x^*) < f((x^* + c)/2)$ (by the definition of $x^*$). Also, by the strict convexity of $f$, we have $f((x^* + c)/2) < (f(x^*) + f(c))/2 < f(c)$. Therefore, we have

$$f(c) > f((x^* + c)/2) > f(x^*). \tag{29}$$

Let $r_1 \in \mathbb{R}$ be such that $f(c) > r_1 > f((x^* + c)/2)$. Now, since $\|f_n - f^*\|_\infty \to 0$ as $n \to \infty$, there exists an positive integer $N$ *s.t.* $|f_n(c) - f(c)| < f(c) - r_1, \forall n \geq N$ and $\epsilon > 0$. Therefore, $f_n(c) - f(c) > r_1 - f(c) \Rightarrow f_n(c) > r_1$. Similarly, we can show that $f_n((x^* + c)/2) > r_1$. Therefore, we have $f_n(c) > f_n((x^* + c)/2)$. Similarly, we can show that $f_n((x^* + c)/2) > f_n(x^*)$. Finally, we obtain

$$f_n(c) > f_n((x^* + c)/2) > f_n(x^*), \quad \forall n \geq N. \tag{30}$$

Now, by the extreme value theorem of the continuous functions, we obtain that for $n \geq N$, $f_n$ achieves minimum (say at $x_p$ in the closed interval $[c, (x^* + c)/2]$. Note that $f_n(x_p) \not< f_n((x^* + c)/2)$ (if so then $f_n(x_p)$ will be a local minimum of $f_n$ since $f_n(x^*) < f_n((x^* + c)/2)$). Also, $f_n(x_p) \neq$

$f_n((x^* + c)/2)$. Therefore, $f_n$ achieves it minimum in the closed interval $[c, (x^* + c)/2]$ at the point $(x^* + c)/2$. This further implies that $x_n^* > (x^* + c)/2$. Therefore, $\liminf_n x_n^* \geq (x^* + c)/2 \Rightarrow c \geq (x^* + c)/2 \Rightarrow c \geq x^*$. This is a contradiction and implies

$$\liminf_n x_n^* \geq x^*. \tag{31}$$

Now consider $g_n(x) = f_n(-x)$. Note that $g_n$ is also continuous and strictly convex. Indeed, for $\lambda \in [0, 1]$, we have $g_n(\lambda x_1 + (1-\lambda)x_2) = f_n(-\lambda x_1 - (1-\lambda)x_2) < \lambda f(-x_1) + (1-\lambda)f(-x_2) = \lambda g(x_1) + (1-\lambda)g(x_2)$. Applying the result from Eq. (31) to the sequence $\{g_n\}_{n \in \mathbb{N}}$, we obtain that $\liminf_n(-x_n^*) \geq -x^*$. This implies $\limsup_n x_n^* \leq x^*$. Therefore,

$$\liminf_n x_n^* \geq x^* \geq \limsup_n x_n^* \geq \limsup_n x_n^*.$$

Hence, $\liminf_n x_n^* = \limsup_n x_n^* = x^*$ $\qquad\square$

## C  EXPERIMENTAL DETAILS

### C.1  HYPERPARAMETER DETAILS

In this section, we outline the tuned hyperparameters for each algorithm on each environment in our experiments. For each algorithm, hyperparameters were tuned over an initial 10 runs with different random seeds. Each algorithm saw the same 10 initial random seeds. For a list of all hyperparameters swept, see Section 5.3. In Table 1, we list the tuned hyperparameters for each algorithm when tuning across continuous-action environments. In Table 2, we list the tuned hyperparameters for each algorithm when tuning across discrete-action environments. In Tables 3, 4, and 5, we list the tuned hyperparameters when tuning per-environment for GreedyAC, VanillaAC, and SAC respectively. Finally, Table 6 outlines the hyperparamters used in the experiments on Swimmer.

| Hyperparameter | $\kappa$ | $\alpha$ | $\tau$ |
|---|---|---|---|
| Greedy Actor-Critic | 1.0 | 1e-3 | 1e-3 |
| Vanilla Actor-Critic | 2.0 | 1e-3 | 1e-3 |
| Soft Actor-Critic | 1.0 | 1e-3 | 1e-3 |

Table 1: Hyperparameters tuned across continuous-action environments for GreedyAC, VanillaAC, and SAC.

| Hyperparameter | $\kappa$ | $\alpha$ | $\tau$ |
|---|---|---|---|
| Greedy Actor-Critic | 10.0 | 1e-3 | - |
| Vanilla Actor-Critic | 1e-1 | 1e-3 | 1e-2 |
| Soft Actor-Critic | 10 | 1e-5 | 10 |

Table 2: Hyperparameters tuned across discrete-action environments for GreedyAC, VanillaAC, and SAC.

| Hyperparameter | $\kappa$ | $\alpha$ | $\tau$ |
|---|---|---|---|
| Acrobot-CA | 1e-1 | 1e-3 | 1e-2 |
| Acrobot-DA | 1e-1 | 1e-2 | - |
| Mountain Car-CA | 1.0 | 1e-3 | 10.0 |
| Mountain Car-DA | 2.0 | 1e-3 | - |
| Pendulum-CA | 1e-1 | 1e-2 | 10.0 |
| Pendulum-DA | 1.0 | 1e-3 | - |

Table 3: Hyperparameters tuned per-environment for GreedyAC.

| Hyperparameter | $\kappa$ | $\alpha$ | $\tau$ |
|---|---|---|---|
| Acrobot-CA | 2.0 | 1e-3 | 1e-3 |
| Acrobot-DA | 1e-1 | 1e-2 | 1e-2 |
| Mountain Car-CA | 2.0 | 1e-3 | 1e-3 |
| Mountain Car-DA | 1.0 | 1e-3 | 1e-2 |
| Pendulum-CA | 1e-2 | 1e-2 | 1e-2 |
| Pendulum-DA | 2.0 | 1e-3 | 1.0 |

Table 4: Hyperparameters tuned per-environment for VanillaAC.

| Hyperparameter | $\kappa$ | $\alpha$ | $\tau$ |
|---|---|---|---|
| Acrobot-CA | 10.0 | 1e-5 | 10.0 |
| Acrobot-DA | 2.0 | 1e-5 | 10.0 |
| Mountain Car-CA | 1.0 | 1e-3 | 1e-3 |
| Mountain Car-DA | 1.0 | 1e-3 | 1e-2 |
| Pendulum-CA | 1e-1 | 1e-2 | 1e-1 |
| Pendulum-DA | 1.0 | 1e-3 | 1.0 |

Table 5: Hyperparameters tuned per-environment for SAC.

| Hyperparameter | $\kappa$ | $\alpha$ | $\tau$ |
|---|---|---|---|
| Greedy Actor-Critic | 1e-2 | 1e-4 | 1e-1 |

Table 6: Hyperparameters Chosen for GreedyAC on Swimmer.

## C.2 NORMALIZATION APPROACH

For each environment, we find the best return achieved by any agent, across all runs, as a simple approximation to a near-optimal return. Table 7 lists these returns for each environment. Then, to obtain a normalized score, we use $1 - \frac{\text{BestValue} - \text{AlgValue}}{|\text{BestValue}|}$, where the numerator is guaranteed to be nonnegative. If AlgValue = BestValue we get the highest value of 1. If AlgValue is half of BestValue, we get $\frac{0.5\text{BestValue}}{|\text{BestValue}|} = 0.5$. If AlgValue is significantly worse than BestValue, the score is much lower.

The AlgValue that we normalize is the point depicted on the sensitivity plot. It corresponds to the Average Return across timesteps and across runs for the algorithm, with that hyperparameter setting in that environment.

For the experiments in Figure 3, where we tune across the complete set of discrete- or continuous-action environments, we first compute the normalized scores just described. Then, we compute the average normalized scores for each algorithm and hyperparameter setting across discrete-action and continuous-action environments separately. We then choose the hyperparameter setting for each algorithm for the discrete- and continuous-action environments based on the hyperparameter setting which resulted in the highest normalized scores. The learning curves for these hyperparameters, for each algorithm, are shown in Figure 3.

| Environment | Continuous | Discrete |
|---|---|---|
| Acrobot | -56 | -56 |
| Mountain Car | -65 | -83 |
| Pendulum | 930 | 932 |

Table 7: Approximate return achieved by an optimal policy. We approximate the return achievable by a near-optimal policy on environment $\mathcal{E}$ by finding the highest return achieved over all runs of all hyperparameters and all agents on environment $\mathcal{E}$.

## C.3 SENSITIVITY PLOTS

We plot parameter sensitivity curves, which include a line for each entropy scale, with the stepsize on the x-axis. Because there are two stepsizes, we have two sets of plots – one for the critic stepsize and one for the actor stepsize. When examining the sensitivity to the critic stepsize, we select the corresponding best actor stepsize. This means that for each point (critic stepsize, entropy scale) = $(\alpha, \tau)$ on the sensitivity plot for the critic stepsize, we find the best actor stepsize and report the performance for that triplet averaged over all 40 runs. We do the same procedure when plotting the actor stepsize on the x-axis, but maximizing over critic stepsize.

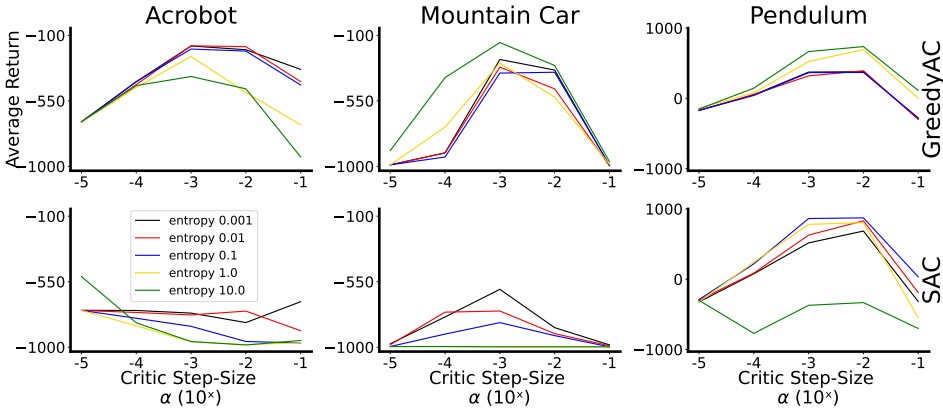

Figure 6: Sensitivity curves for the critic step-size hyperparameter $\alpha$ for GreedyAC and SAC, with one line for each entropy scale tested. The critic step-size is plotted on a logarithmic scale on the x-axis.

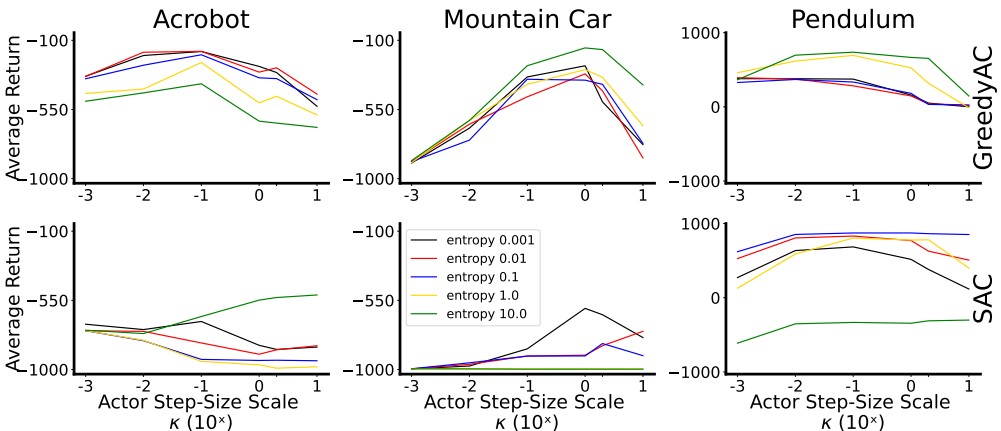

Figure 7: Sensitivity curves for the actor stepsize scale hyperparameter $\kappa$ for both GreedyAC and SAC, with one line for each entropy scale tested. The actor step-size scale is plotted on a logarithmic scale on the x-axis.

# D  ABLATION STUDY ON SAC

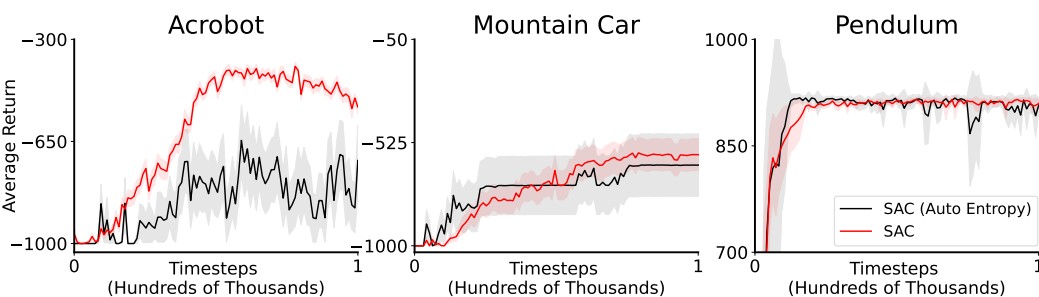

Figure 8: Learning curves over 40 runs for SAC and 10 runs for SAC (Auto Entropy) with shaded regions denoting standard error. The entropy scale for SAC as well as the entropy scale step-size for SAC (Auto Entropy) are **tuned using a grid search**.

Modern variants of SAC utilize a trick to automatically adapt the entropy scale hyperparameter during training (Haarnoja et al., 2018b). In order to gauge which variant of SAC to use in this work, we performed an ablation study where we studied SAC with and without automatic entropy tuning. We ran SAC with automatic entropy tuning for 10 runs. Hyperparameters were swept in the same sets as listed in Section 5.3. Additionally, we swept entropy scale step-sizes $\beta = 10^z$ for in $z \in \{-4, -3, -2\}$ for automatic entropy tuning. Figure 8 shows the learning curves of SAC with automatic entropy tuning, over 10 runs, and SAC without automatic entropy tuning over the 40 runs conducted for the experiments in the main text. As can be seen in the figure, performing a grid search over the entropy scale hyperparameter never degrades performance compared to using automatic entropy tuning, and in some cases results in better performance than when using automatic entropy tuning. Because of this, we decided to use manual entropy tuning through a grid search in our experiments, which also allows us to characterize the sensitivity of SAC's performance with respect to the entropy scale hyperparameter.

