# OpenReview forum: "Greedy Actor-Critic: A New Conditional Cross-Entropy Method for Policy Improvement"
_ICLR.cc/2023/Conference — ICLR 2023 poster_

### Official Review · Reviewer_y7Rc · 2022-10-23

**Confidence:** 3
**Correctness:** 4
**Technical Novelty And Significance:** 3
**Empirical Novelty And Significance:** 2
**Recommendation:** 6

**Clarity, Quality, Novelty And Reproducibility:**

**Clarity**: at a high level good, but have some concerns for some details.

**Quality**: sound as far as I can tell.

**Novelty**: good.

**Reproducibility**: only concern I have is due to clarity concerns for some technical details, as mentioned above.

---

**Detailed Comments**:
- p.2: "attempt to obtain (unbiased) estimates of this gradient" --> what is "this gradient"? I think you mean "the gradient of this objective"?
- Third paragraph of Section 2: suddenly a boldface $\boldsymbol{w}$ is introduced as subscript for a policy $\pi$. Earlier in the paper, $\theta$ was used to denote the parameters of a policy $\pi_{\theta}$. Why did this suddenly change to $\boldsymbol{w}$? Is it the same thing, or something else? Without explanation, I would assume it was just the same thing... but later in the paper I keep seeing unexplained mixes of $\boldsymbol{w}$, $\theta$, and even yet again a new $\boldsymbol{w}'$. I can guess that that last one is used for the proposal policy, due to how it is used in the update that also involves entropy regularization... but I really shouldn't have to guess this, this should be clearly explained. In the later parts I think $\theta$ is more commonly used only as subscript for $Q$, but (especially early in the paper) also for policies, so... I continue to be confused throughout all of the paper.
- p.3: "for $h < N$" --> I think this should be $h \leq N$?
- p.3: "narrows around higher valued $\theta$" --> technically this can be read as saying that it is simply trying to make the values of the parameters $\theta$ high. I think you rather mean that it narrows around parameters $\theta$ that lead to policies that achieve high returns.
- "CEM, however, finds the single-best set of optimal parameters" --> does it really find the optimal parameters? Or does it just try to approximate them?
- Figure 1 has equations saying that policies are equal to certain gradients. Are they really equal to the gradients? Or are their parameters just updated according to those gradients?
- p.4: "The actor $\boldsymbol{w}_t$" / "The proposal $\boldsymbol{w]_t'$" --> do these symbols actually denote the full actor/proposal [policy]? Or just their parameters?
- At the end of page 4, I do not understand why a proposal policy is no longer necessary for problems with discrete action spaces, and why it would no longer be useful to sample a distribution of actions. To me, exactly the same intuition would still apply. I think it would be useful to provide more explanation here.


**Other Minor Comments**:
- p.3: "are sample from" --> are sampled from
- Caption Fig 1: "more quickly that" --> more quickly than
- The legends for Figures 2 and 3 need to be slightly bigger to be readable.
- p.9: SOTA --> I know what the acronym means, but it is never fully written out and should not be used without fully writing out at least once.

**Strength And Weaknesses:**

**Strengths**:
- While the paper is rather technical and dense, in general it is possible to follow along with the main ideas.
- Theoretical and algorithmic contributions appear to be sound, with good emprical results.

**Weaknesses**:
- In several parts of the more technical writing, it seems that there are slight mistakes and confusing notation (notation suddenly changes or is not explained at all), making some parts difficult to follow (see detailed comments below).
- There are some statements where I'd appreciate a bit more intuition/explanations (see detailed comments below).

---

**After discussion**: the revision of the paper has fixed and clarified many of my concerns, and I have updated my score.

**Summary Of The Paper:**

This paper proposes the Greedy Actor-Critic approach, which uses two policies, and a new CCEM approach: a conditional (on states) varianta of CEM. The critic is trained using Sarsa updates (with a replay buffer). A *proposal policy* (one of the two policies) is used to sample a distribution of actions, which are then evaluated and ranked by the critic. The top N of these actions are used for training the policies. The main policy (the second of the two policies) is used to select actions to be executed in the environment. Both policies are updated such that they increase the likelihood of the top N actions mentioned previously, but the proposal policy additionally has an entropy regularization term (which the main policy doesn't).

**Summary Of The Review:**

Potentially a good paper, but currently having slightly too many issues with imprecise statements and confusing / unexplained notation.

---

**After discussion**: the revision of the paper has fixed and clarified many of my concerns, and I have updated my score.

---

> ### Author Response · Authors · 2022-11-11
> **Response to Reviewer y7Rc**
>
> We thank the reviewer for the time put into this thorough review. Incorporating your suggestions has improved the paper.
>
> Thank you for catching several typos that have caused confusion. We have now updated the paper with all of these fixed. To clarify here about some of the changes that caused more confusion:
>
> 1. In the second section, the parameters \theta are for the critic, while the parameters w are for the actor policy, and the parameters w’ are for the proposal policy. We incorrectly used a mix of \theta and w for the parameters of the actor policy.
>
> 2. On page 3, we first simply describe the original CEM update used to optimize a single, stationary function. This is why we state that the distribution p_t narrows around higher valued \theta under values assigned by some function f(\theta). In the case of the CCEM (not CEM), the policy narrows probability around maximal actions, with the weights adjusted to achieve this. We see that this paragraph should have better clarified that it describes CEM rather than CCEM; we have now fixed this and used a different parameter name for CEM (beta) so it does not clash with our parameters.
>
> 3. When we said “CEM, however, finds the single-best set of optimal parameters" we really meant "CEM, however, *attempts* to find the single-best set of optimal parameters". We absolutely agree that it only approximates in most cases. This is now fixed.
>
> 4. For figure 1, the actor and proposal policies are updated with the stated gradients and are not equal to these gradients, as the reviewer has correctly pointed out. This has also been addressed in the figure. On page 4, the reviewer is correct that w_t and w_t’ are the parameters of the actor and proposal policies, and do not represent the complete policies.
>
> Finally, let us address your question about discrete-action CCEM. You are right that we could use exactly the same approach for discrete actions as continuous. And in fact, for a very large discrete action space, we should! However, if the action space is small (e.g., 10), then there is no point in sampling actions. In all of our experiments, the number of actions in the discrete-action setting is small enough (3 or 4) that the top percentile for any rho we chose was actually just the top action. For this reason, we chose to write this simpler discrete-action algorithm. We have now changed the pseudocode to say “if actions discrete and |A| < 1/rho” and explained more in the paragraph below. Thank you for this question that helped clarify the algorithm, and avoid making the discrete-action case overly specific to small discrete action spaces.

---

### Official Review · Reviewer_BG9i · 2022-10-24

**Confidence:** 4
**Correctness:** 4
**Technical Novelty And Significance:** 2
**Empirical Novelty And Significance:** 3
**Recommendation:** 8

**Clarity, Quality, Novelty And Reproducibility:**

- Significance: very high. The results presented in this paper are very strong, and the approach solves a problem (continuous-action Conservative Policy Iteration)
- Novelty: moderate. There is a whole field of RL in which the actor imitates some (greedy) function of the critic, and this work builds on/reinvents parts of that.
- Calrity: high. The paper is well-written and easy to follow.

**Strength And Weaknesses:**

Strengths:

- The algorithms is simple, elegant and performs very well even on challenging tasks (Atari games, Mujoco tasks)
- The paper is clear and well-written

Weakness:

My use of "pursue" in "Summary of the paper" is on purpose. This paper actually presents an algorithm for continuous-action Conservative Policy Iteration [1], which can also be seen as continuous-action state-dependent Pursuit [2]. There is a large and exciting field of Reinforcement Learning that considers gradient-free optimization of the actors, using imitation learning to pursue the greedy policy of critics. Such work include:

- The Actor-Mimic [3], in which a DQN policy is distilled into an actor
- Dual Policy Iteration [4], the formalism behind AlphaZero, in which the actor pursues a proposal policy (in the case of MuZero, obtained by planning with MCTS)
- Approximate Conservative Policy Iteration [1], in which the proposal policy is the greedy policy of an on-policy critic. This paper, with discrete actions (Algorithm 2, discrete), is Approximate Conservative Policy Iteration
- Bootstrapped Dual Policy Iteration [5], close to Approximate Policy Iteration, but that shows that off-policy critics (learned with Q-Learning instead of SARSA) work, leading to extremely high sample-efficiency and hyper-parameter robustness (claims also made in this paper, and mostly due using Dual Policy Iteration instead of Policy Gradient)
- Batch/Offline Reinforcement Learning also has extensive literature on proposal policies and how to sample actions. For instance, Batch-Constrained Q-Learning [6] learns the proposal policy using a conditional Variational Autoencoder. This work is however only distantly related to this paper.

To my knowledge, the proposal policy presented in this paper, and its use to compute approximate greedy policies with continuous actions, is novel. The fact that it works is however not surprising, as the literature mentioned above consistently, over the past years, showed that doing something else that Policy Gradient for actor-critic training is very good indeed.

My points above lead to two remarks:

- Would it be possible for the authors to have a look at the world of Conservative Policy Iteration, Dual Policy Iteration and Policy Distillation, and briefly discuss that in the paper?
- Why is the critic learned with SARSA in this paper? [6] shows that using SOTA critic-learning algorithms (the proposed algorithm combines twin-delayed critics (TD3) with Boostrapped DQN) leads to significant gains in sample-efficiency. The only challenge is that all these learning algorithms assume the existence of $\argmax_a Q(s, a)$, so, discrete actions, but this paper happens to propose a way to compute that max and argmax using CEM and a proposition policy.

[1]: Kakade, S., Langford, J.: Approximately optimal approximate reinforcement learning. In: International Conference on Machine Learning (ICML). pp. 267–274 (2002)

[2]: Agache, M., & Oommen, B. J. (2002). Generalized pursuit learning schemes: New families of continuous and discretized learning automata. IEEE Transactions on Systems, Man, and Cybernetics, Part B (Cybernetics), 32(6), 738-749.

[3]: Parisotto, E., Ba, J. L., & Salakhutdinov, R. (2015). Actor-mimic: Deep multitask and transfer reinforcement learning. arXiv preprint arXiv:1511.06342.

[4]: Sun, W., Gordon, G. J., Boots, B., & Bagnell, J. (2018). Dual policy iteration. Advances in Neural Information Processing Systems, 31.

[5]: Steckelmacher, D., Plisnier, H., Roijers, D. M., & Nowé, A. (2019, September). Sample-efficient model-free reinforcement learning with off-policy critics. In Joint European Conference on Machine Learning and Knowledge Discovery in Databases (pp. 19-34). Springer, Cham.

[6]: Scott Fujimoto, David Meger, and Doina Precup. 2019. Off-policy deep reinforcement learning without exploration. In International Conference on Machine Learning. 2052–2062

**Summary Of The Paper:**

The paper proposes an Actor-Critic algorithm in which the actor learns to pursue (imitate) the greedy policy of the critic, using, with continuous actions, a method inspired from the Cross-Entropy method (pursuing the top-K actions from a sample of relatively good actions). With continuous actions, the core contributions of the paper are the suggestion of using CEM to approximate $\argmax_a Q(s, a)$, and the introduction of an entropy-regularized proposal policy $\pi_{w'}$, to choose which actions to sample for computing the top-K of $Q(s, a)$.

**Summary Of The Review:**

Very good idea, but learning the critic with SARSA? It seems that it helps with the proof, but maybe not the empirical performance. Other algorithms that train the actor towards target actions (as opposed to using Policy Gradient) should be discussed in the paper.

---

> ### Author Response · Authors · 2022-11-11
> **Response to Reviewer BG9i**
>
> We thank the reviewer for the insightful review and for pointing out this related line of research.
>
> First, thank you for these useful citations that we missed! Our plan is to add a few paragraphs, connecting to this literature on conservation policy iteration and dual policy iteration. (Explained more below).
>
> Let us start with the points about approximate policy iteration (API) and conservative policy iteration (CPI). We agree that our approach is related to API and CPI. We start the paper pointing out research that examines relationships between AC and API approaches (Veillard, et al. and Chan et al.), and then directly state the connection in the background at the top of page 3. In an earlier iteration of the paper, we actually started the paper discussing the utility of viewing policy optimization through a policy iteration lens, rather than as policy gradients. However, we decided to avoid this bigger story and keep it more focused on the actor update, and particularly contrasting to the SAC update. We absolutely agree that our Greedy AC algorithm can be seen as API, and your point about it being like CPI is also reasonable (we do change the actor slowly, rather than setting it to the greedy policy right away). Practical implementations of CPI continue to be investigated (e.g., Deep CPI), as the original CPI uses interpolation between policies (rather than parameters). Greedy AC could be seen as another attempt at a practical implementation; we will discuss this connection to CPI.
>
> A clarification question for you: The Deep CPI approach minimizes a KL to approximate the interpolation proposed in CPI; but, as far as we know, it does not yet have the guarantees of CPI. If you feel Greedy AC has a stronger connection to CPI, then we would love to hear your thoughts about this more. As it stands, we are not sure why you say our discrete action algorithm is exactly CPI with a proposal policy that is greedy.
>
> The second point was about the connection to Dual Policy Iteration (DPI) and other such imitation learning approaches like Actor-Mimic. These approaches do look related: DPI has two policies, one which is guiding the other. For example, one policy might be an expensive tree search and another a learned neural network, trained to mimic the first (expert or guide) policy. CCEM, on the other hand, uses two policies differently. Our actor does not imitate our proposal policy. Rather, the proposal policy is used to improve the search over the nonconcave surface of Q. It samples actions more broadly, to make it more likely to find a maximizing action. Further, the actor increases the likelihood of only the top actions and does not imitate the proposal policy. It is trying to match the percentile greedy policy, as a target policy.
>
> Despite the difference in goals, the algorithms can look quite similar and importantly both methods (DPI and Greedy AC) are not using a PG update. It is warranted to properly place our algorithm relative to those. The BDPI paper uses an Actor-Mimic update (eq 5) that has similarities to our maximum likelihood update. This literature on policy optimization methods is vast, and there are nuanced relationships between them. We will add a more in-depth section to the appendix explaining connections to these methods, including CPI and policy distillation. We are working on this section now but wanted to post a response here sooner; we will upload a new revision and message here before the end of the rebuttal period with this new section. If you have any further concerns about properly connecting and placing our work relative to this literature, then we are happy to discuss further.
>
>  As for the second comment about Sarsa and Q-learning: under continuous actions, it is difficult to use Q-learning. We can use the greedy action from our actor (easy to get), rather than an action sampled from the actor. However, this is still not precisely Q-learning since this actor’s greedy action may not actually correspond to max_a’ Q(s’,a’). Rather, using the greedy action from the actor is like off-policy Sarsa where the target policy is the greedy actor. If this is your proposal, then actually we agree that this is a sensible choice (and applies to both discrete and continuous actions). We did actually run the algorithm this way originally, but found little difference between using off-policy Sarsa and Sarsa. For this reason we chose to keep the approach as similar to standard actor-critic algorithms and only change the key component: the actor update. The paper you cited motivates further that we should re-investigate this question. And, in general, one of our key next steps was to ask: how should we improve the critic, for this actor update?

---

> > ### Comment · Reviewer_BG9i · 2022-11-16
> > **Very interesting details**
> >
> > Thank you for the thorough discussion of my remarks, I really appreciate it.
> >
> > Due to space constraints and in the interest of time, I wrote some elements of my review in a somewhat simplified way. For instance, I don't touch on the difference between interpolating between policies, and policy parameters. I welcome your discussion in that regard, and your potential inclusion of it in the paper, as it is an important point that is easy to overlook. With Deep RL, we indeed only have access to policy parameters. It is therefore impossible to implement the CPI update rule that directly modifies $\pi(s, a)$. My comment on "is exactly CPI" is therefore a simplification: your loss invites the actor to increase the probability of good actions, similar to what CPI intuitively does. This is indeed pointing in the same direction as the Actor-Mimic loss, or the Mean-Squared-Error loss used in BDPI. Just various expressions of "actions that the critic likes should be done more often".
> >
> > Regarding Q-Learning, I was referring to the idea of Batch-Constrained Q-Learning where actions are sampled (in their case from a VAE, not a proposal distribution), then the max and argmax are computed on them, and the result of that is used as an approximation of $\max_{a'} Q(s', a')$. I think that this operation may lead to higher Q-Values, and potentially faster learning, than simply using $Q(s', \pi(s'))$. It can however become unstable, and introduces the problem of over-estimating the Q-Values. This brings me to the improvement of the critic, where, if we assume them a bit more Q-Learning-like and a bit more off-policy, becomes amenable to:
> >
> > - Experience Replay, maybe even Prioritized Experience Replay (probably the most important aspect for sample-efficiency if learning remains stable)
> > - Dueling architectures, maybe even also auxiliary tasks (but auxiliary tasks do not seem to be done that often anymore in RL)
> > - Clipped DQN as in the TD3 paper (now I don't remember anymore if you already do it with SARSA)
> > - Bootstrapped DQN, which allows to produce mean and variances of Q-Values instead of just a single (mean) prediction. Then, a sort of confidence interval could be used when sorting the actions by Q-Value, instead of sorting according to their mean, to do smart things such as: encourage the actor to execute actions with high-certainty Q-Values, and encourage the guide to select actions that have low-certainty Q-Values

---

> > > ### Author Response · Authors · 2022-11-19
> > > **Thank you for the follow-up**
> > >
> > > Thank you for the clarification!
> > >
> > > We have now added the literature you mentioned to our paper, with Appendix A explaining the connections.
> > >
> > > We absolutely see your point that it could be useful to incorporate some of the sample efficiency improvements underlying DQN and follow-up algorithms (like dueling networks and bootstrap DQN). We can (and do) use some of these improvements already in off-policy Sarsa. For example, our critic in Greedy AC uses an off-policy Sarsa update with ER. We can still replay transitions (s,a,s’,r), but a’ is selected using pi rather than using a maximum. We could, for example, improve this using prioritized ER and even a second value function (dueling or advantage estimation).
> > >
> > > Nonetheless, this might be slightly besides your point. Increasing action-values more and reasoning about the best action, rather than reasoning about returns for our current suboptimal policy, does seem like it should be more efficient. This is something we plan to explore more fully next. In fact, we had previously tested using an update that is more like Q-learning: starting an optimization from the action given by our actor and then doing a few steps of gradient ascent to get closer to the maximizing action. At the time, we found that this was not notably different from Sarsa, and so decided to use the simpler Actor-Critic update (start simpler where possible). As we said above, though, this was preliminary and more needs to be done here! Thank you for highlighting that this is something we should investigate further, and further highlighting that we can try to leverage some of the algorithmic optimizations designed for Q-learning algorithms.

---

### Official Review · Reviewer_zQ75 · 2022-11-02

**Confidence:** 4
**Correctness:** 3
**Technical Novelty And Significance:** 3
**Empirical Novelty And Significance:** 2
**Recommendation:** 6

**Clarity, Quality, Novelty And Reproducibility:**

**Clarity**:
The writing is clear in most places. Here are some places that could be improved:
- Regarding the convergence analysis (cf. Theorem A.1), only informal results are provided in the main text. While this arrangement could provide intuition, on my first read, it is a bit confusing to me what kind of convergence result has been established in the paper. I would suggest that the authors at least provide a formal statement of Theorem A.1 as this result appears to be one of the major components of the paper.
- The caption of Figure 1 is somewhat confusing. It would be helpful to clarify the description of Figure 1.
- It is not clear how to parametrize the percentile-greedy policy by some parameter $w$.

**Novelty**

Please see the above comments about the weaknesses and the missing references.

**Reproducibility**

As mentioned above, the performance of the baseline method are somewhat different from the existing public benchmarks.


**Strength And Weaknesses:**

**Strength**
- This paper first introduces a new perspective of applying CEM to RL and thereby proposes the GreedyAC algorithm with theoretical convergence and improvement guarantee.
- The proposed method is not sensitive to the entropy regularization parameter and appears to have some empirical advantage over the popular benchmark SAC.

**Weaknesses**
- While the theoretical results are interesting and could have their own merits, the algorithm analyzed in theory appear to deviate from the actual algorithm introduced in the paper (cf. Algorithms 2-3).
- There are several concerns about the experimental results:
    - SAC: The reported empirical performance of SAC looks quite different from the existing benchmarks. For example, from Figure 2, it is surprising that SAC fails to solve the simple Acrobot and MountainCar tasks. By contrast, in the existing benchmarks (e.g., Table 1 in https://arxiv.org/abs/1907.02057), SAC appears quite strong in these classic tasks of OpenAI gym. It is not clear what makes the difference, and further explanation is needed.
    - Sensitivity: One of the claims is that GreedyAC empirically could be less sensitive to the entropy regularization parameter than SAC. However, it is not clear why this is true from the design or the convergence analysis. Further justification is needed for this claim, either from intuition, theoretical analysis or ablation studies.
- Some components of GreedyAC look somewhat ad hoc and would require more explanation. For example, the proposal distribution plays an important role in the policy improvement of GreedyAC, and it relies on an entropy regularization term.
- Several missing references on CEM for RL: There are several prior works on the combination of CEM and value-based methods, such as [Simmons-Edler et al. 2018] and [Shao et al. 2022]. These references are missing in this paper.

[Simmons-Edler et al. 2019]  Riley Simmons-Edler, Ben Eisner, Eric Mitchell, Sebastian Seung, Daniel Lee (2019). “Q-Learning for Continuous Actions with Cross-Entropy Guided Policies”. In The Reinforcement Learning for Real Life (RL4RealLife) Workshop in the 36th International Conference on Machine Learning (ICML), 2019.

[Shao et al., 2022] Lin Shao, Yifan You, Mengyuan Yan, Shenli Yuan, Qingyun Sun, Jeannette Bohg, "GRAC: Self-Guided and Self-Regularized Actor-Critic." Conference on Robot Learning. PMLR, 2022.



**Summary Of The Paper:**

This paper proposes a new policy improvement scheme called conditional cross-entropy method (CCEM), which leverages the cross-entropy method in global optimization to keep track of the set of maximal actions. The proposed algorithm is a variant of the actor-critic methods and called GreedyAC, which (i) uses CCEM for updating the actor and (ii) applies entropy regularization on the loss function of the actor. The proposed method enjoys the advantages of (i) being non-sensitive to the entropy regularization parameter, (ii) concentrating on the greedy policy more slowly, (iii) having better empirical results in some environments than SAC, and (iv) enjoying policy improvement guarantee and convergence.

**Summary Of The Review:**

This paper presents a new perspective of applying CEM to policy improvement. While the theoretical result is interesting, there are some issues with whether the analysis actually captures the behavior of the proposed algorithm as well as with the empirical evaluation of the baselines and claim on sensitivity. Moreover, the novelty of this paper should be further clarified as there are some missing references that look relevant but not discussed in this paper.

========== Post-rebuttal ==========
I would like to thank the authors for providing a very detailed response. My main concerns, the difference between the algorithm and the analysis as well as the experimental results of the benchmark methods, have been addressed. As a result, I decided to raise my score from 5 to 6.

---

> ### Author Response · Authors · 2022-11-11
> **Response to Reviewer zQ75**
>
> We thank the reviewer for the detailed feedback and hope that our clarifications made below can address the reviewers concerns and raise the reviewer’s rating of our paper.
>
> Our GreedyAC algorithm (Algorithms 2-3 in the paper) does deviate from the theoretical algorithm analyzed in the paper as reviewer zQ75 has pointed out. Although this is a limitation, we would like to point out that nearly all algorithms in reinforcement learning also suffer from this limitation. For example, many popular actor-critic algorithms deviate from the relevant theory presented in their papers [1, 2, 3]. The analyzed algorithm has similar underlying principles (a slower changing proposal policy, a maximum likelihood actor update, etc), but makes certain assumptions to facilitate theoretical analysis. We do acknowledge this in the work and highlight that a key next step is to extend this analysis to better reflect the algorithms we use in practice. We do think, though, that the theory given her facilitates that next step.
>
> Although SAC is a strong benchmark algorithm on Mujoco domains, we are unaware of published works which benchmark SAC on the classic variants of Mountain Car, Pendulum, and Acrobot, including the paper referenced by reviewer zQ75 [7]. We would like to stress that in our empirical study, we used the classic variants of Mountain Car, Pendulum, and Acrobot (see the in-text descriptions along with the corresponding references in the paper) which are different from (and more difficult than) the OpenAI Gym variants [5] and are difficult for deep reinforcement learning algorithms to solve [6]. For example, the OpenAI variants of these classic domains generally have generous start states, aggressive episode cutoffs, and nicely shaped reward functions. Furthermore, the preprint paper cited by reviewer zQ75 modifies the reward function of many of these environments to make them easier to solve (see section 4.1 and appendix A) [7]. For example on Mountain Car they provide a reward equal to the position of the car, whereas we provide a reward of -1 per step. Their version is easier due to a nicely shaped reward. The cited preprint [7] also uses different state observations from those used in our paper (once again, see the in-text descriptions and the corresponding references in our paper). Furthermore, we have tested our SAC implementation on OpenAI Gym’s HalfCheetah and were able to reproduce the results found in [1].
>
> The experiments conducted in the preprint cited by reviewer zQ75 ([7]) also use very few seeds (a total of 4) for experimental repetition. This brings into question the experimental results reported in [7]. We have used a total of 40 seeds to evaluate SAC on the classic control domains, providing a better estimate of the true mean performance of SAC on these problems than those reported in [7].
>
> Our claim that GreedyAC is less sensitive to entropy is justified empirically and also we provide intuition that its role inside only the proposal policy should result in reduced sensitivity. It has been repeatedly reported that when using entropy-regularization in the actor the entropy can be difficult to choose; we remove entropy-regularization from the actor. Entropy for us only plays the role of making the proposal policy narrow more slowly than the actor. Entropy-regularization in the actor affects exploration, the final solution and even the loss surface. We do not fully understand the request for an ablation study to better show reduced sensitivity, but would like to understand this comment if you could further clarify.
>
> The reviewer states that choices in GreedyAC are adhoc. Beyond the proposal policy, we are not sure which choices were adhoc and again would much appreciate more detail about what you mean. As for the proposal policy, you are right that different approaches could be used to make it concentrate more slowly than the actor. We chose to use entropy-regularization for this, to provide a more clear apples-to-apples comparison with SAC. I would not call it an adhoc choice, rather it was a deliberate and reasonable choice. We can add a discussion that other options might be possible (for example, we are currently experimenting with using a larger percentile for the proposal policy as compared to the actor to get the same effect).
>
> Thank you for these references! We have now added them to the footnote discussing the two other uses of CEM in RL, on page 2. These both extend QT-Opt, which runs the CEM algorithm on Q(s,dot) to get the maximal action. They introduce explicit policies that use this maximal action, but still rely on fully running CEM from scratch each time they want to find max+a’ Q(s’,a’).
>
> (continued in next thread...)

---

> > ### Author Response · Authors · 2022-11-11
> > **Continued..**
> >
> > For the last three clarification comments:
> > 1. The theorem statement is quite technical, as is often the case for RL convergence results. We did not have space to give the necessary preamble to introduce it fully. An informal statement with a summary of the proof was the most we could fit.
> > 2. We have fixed this caption.
> > 3. In fact, it is hard to parameterize the percentile greedy policy! Which is why we do not. Instead, the CCEM update moves towards this percentile greedy policy without explicitly having to represent it.
> >
> > We hope these clarifications have addressed your concerns. If not, then we are more than happy to address further comments.
> >
> > [1] Soft Actor-Critic: Off-Policy Maximum Entropy Deep Reinforcement Learning with a Stochastic Actor. Tuomas Haarnoja, Aurick Zhou, Pieter Abbeel, Sergey Levine. International Conference on Machine Learning, 2018.
> > [2] Continuous Control with Deep Reinforcement Learning. Timothy P. Lillicrap, Jonathan J. Hunt, Alexander Pritzel, Nicolas Heess, Tom Erez, Yuval Tassa, David Silver, Daan Wierstra. International Conference of Learning Representations, 2016.
> > [3] Greedification Operators for Policy Optimization: Investigating Forward and Reverse KL Divergences. Alan Chan, Hugo Silva, Sungsu Lim, Tadashi Kozuno, A. Rupam Mahmood, Martha White. ArXiv Preprint, 2022.
> > [4] Reinforcement Learning: An Introduction. Richard Sutton and Andrew Barto. MIT Press, 2018.
> > [5] OpenAI Gym. Greg Brockman and Vicki Cheung and Ludwig Pettersson and Jonas Schneider and John Schulman and Jie Tang and Wojciech Zaremba. ArXiv Preprint, 2016.
> > [6] Revisiting Rainbow: Promoting more Insightful and Inclusive Deep Reinforcement Learning Research. Johan S. Obando-Ceron, Pablo Samuel Castro. International Conference on Machine Learning, 2021.
> > [7] Benchmarking Model-Based Reinforcement Learning. Tingwu Wang, Xuchan Bao, Ignasi Clavera, Jerrick Hoang, Yeming Wen, Eric Langlois, Shunshi Zhang, Guodong Zhang, Pieter Abbeel, Jimmy Ba. ArXiv Preprint, 2019.

---

### Decision · Program_Chairs · 2023-01-20

**Decision:**

Accept: poster

**Justification For Why Not Higher Score:**

The methods, while simple and novel and reproducible, are not surprising given existing the literature.

**Justification For Why Not Lower Score:**

The methods do make a contribution that should appear in the literature.

**Metareview: Summary, Strengths And Weaknesses:**

One reviewer was quite excited by the paper to the point where they replicated the results.  Quoting from that review:

This paper presents an algorithm for continuous-action Conservative Policy Iteration which can also be seen as continuous-action state-dependent Pursuit.  To my knowledge, the proposal policy presented in this paper, and its use to compute approximate greedy policies with continuous actions, is novel.

A couple initial reviews were negative based on disappointments in the quality of the paper but scores were later raised in response to author feedback and improvements in the paper.  While one was excited, they also said:

The fact that [this] works is however not surprising, as existing literature consistently, over the past years, showed that doing something other that Policy Gradient for actor-critic training is very good indeed.

The novelty of the work and the reproducibility of the results were, in the end, judged to warrant publication.



**Note From Pc:**

if the above contains the word "oral" or "spotlight" please see: "oral" presentation means -> notable-top-5% and "spotlight" means -> notable-top-25%. As stated in our emails, we are disassociating presentation type from AC recommendations

**Summary Of Ac-Reviewer Meeting:**

The enthusiasm of the one strongly positive reviewer was clear in the meeting and other reviewers did not object.  The strongly positive reviewer gave an enthusiastic summary of the paper.